# Multi-Token Prediction Needs Registers

Anastasios Gerontopoulos[1,3]    Spyros Gidaris[2]    Nikos Komodakis[1,3,4]

[1]Archimedes, Athena Research Center    [2]valeo.ai
[3]University of Crete    [4]IACM–Forth

## Abstract

Multi-token prediction has emerged as a promising objective for improving language model pretraining, but its benefits have not consistently generalized to other settings such as fine-tuning. In this paper, we propose `MuToR`, a simple and effective approach to multi-token prediction that interleaves learnable register tokens into the input sequence, each tasked with predicting future targets. Compared to existing methods, `MuToR` offers several key advantages: it introduces only a negligible number of additional parameters, requires no architectural changes—ensuring compatibility with off-the-shelf pretrained language models—and remains aligned with the next-token pretraining objective, making it especially well-suited for supervised fine-tuning. Moreover, it naturally supports scalable prediction horizons. We demonstrate the effectiveness and versatility of `MuToR` across a range of use cases, including supervised fine-tuning, parameter-efficient fine-tuning (PEFT), and pretraining, on challenging generative tasks in both language and vision domains. Our code is available at https://github.com/nasosger/MuToR.

## 1 Introduction

Autoregressive Transformer architectures have become the cornerstone of modern Large Language Models (LLMs), enabling unprecedented capabilities across a wide range of natural language processing tasks [Achiam et al., 2023, Liu et al., 2024]. Their success has also extended to domains such as image generation [Esser et al., 2021, Sun et al., 2024] and multimodal models [Alayrac et al., 2022, Liu et al., 2023]. These models are primarily trained using a simple yet effective approach: next-token prediction with teacher forcing. By supplying ground truth tokens as context, teacher forcing enables fully parallelized computation via masked self-attention, thus accelerating and stabilizing training.

However, next-token prediction with teacher forcing has notable limitations. Models trained this way often focus on short-term patterns while struggling with harder, long-range decisions. Prior work has shown that this setup can lead to shortcut learning, where valuable supervision diminishes [Bachmann and Nagarajan, 2024], and that it underperforms on tasks requiring planning [Bubeck et al., 2023]. These findings strongly suggest the need for training objectives that go beyond standard next-token prediction.

To address these limitations, multi-token prediction training [Qi et al., 2020, Gloeckle et al., 2024, Liu et al., 2024] has emerged as a promising alternative. Rather than predicting just one token at a time, the model learns to predict multiple future tokens at each position. Recent implementations achieve this through additional transformer output heads: Gloeckle et al. [2024] employs parallel heads, one for each future token position, while Liu et al. [2024] uses sequential heads. Crucially, this approach is used only during training, as its primary goal is to provide a more informative learning signal rather than to speed up inference. Compared to standard teacher forcing, multi-token prediction encourages the model to develop internal "planning" representations, and mitigates overfitting to local patterns. Experiments by Gloeckle et al. [2024] demonstrate that it leads to improved generative performance and increased data efficiency.

39th Conference on Neural Information Processing Systems (NeurIPS 2025).

*Motivated by these findings, we explore whether a more effective approach for multi-token prediction can further enhance autoregressive transformers.*

We propose a simple but powerful modification: instead of adding extra transformer layers for future token prediction, we introduce *register tokens*—special tokens interleaved between the regular tokens. Each register token is assigned a randomly sampled offset $d$, and the model is trained to predict the token $d$ steps ahead (rather than just the next token) for these tokens. The original next-token prediction objective remains unchanged for all regular tokens.

These register tokens are used exclusively during training to propagate a richer supervisory signal through the model. At inference time, they are discarded to preserve generation speed. This is made possible by a carefully designed attention mask: register tokens are allowed to attend only to preceding regular tokens—enabling them to learn predictive representations—while regular tokens are entirely blind to register tokens in both directions. This ensures full compatibility with standard autoregressive inference while encouraging the model to internalize forward-looking, multi-step planning during training.

Compared to approaches that add output heads or transformer blocks, our register-based method offers several key advantages:

**No architectural changes:** Only a small number of additional trainable parameters are introduced through register embeddings, while the core transformer layers remain untouched and no extra output heads are required.

**Fine-tuning compatibility:** Our method is particularly well-suited for fine-tuning pretrained LLMs (e.g., Llama [Grattafiori et al., 2024], Gemma [Gemma Team et al., 2024]). It introduces minimal parameter overhead, preserves original attention patterns for regular tokens, and uses carefully selected position ids and attention masks for register tokens—bringing multi-token prediction closer to the next-token pretraining setup. In contrast, previous methods [Gloeckle et al., 2024, Liu et al., 2024] rely on separate transformer heads, adding many new parameters that must be trained from scratch, making them less effective in fine-tuning scenarios.

**Scalable prediction horizons:** Because the number of register tokens remains fixed regardless of the offset $d$, the training cost is independent of the prediction horizon, which can be scaled arbitrarily. Register tokens thus provide greater flexibility in future token prediction. For example, in autoregressive image generation, our method naturally extends to predicting tokens in a two-dimensional neighborhood—a capability not easily achieved by adding output heads.

Overall, our work delivers the following contributions:

- We introduce `MuToR` (Multi-Token prediction with Registers), a novel multi-token prediction method that employs trainable, interleaved register tokens tasked with predicting multiple future targets. `MuToR` enables scalable prediction horizons with minimal additional parameters and seamlessly integrates with any pretrained autoregressive language model without architectural modifications.

- Through experiments on language modeling benchmarks, we validate the effectiveness of `MuToR` in both supervised fine-tuning and parameter-efficient fine-tuning (PEFT) settings, consistently surpassing standard fine-tuning baselines under equivalent training compute.

- We further demonstrate the versatility of `MuToR` by applying it to autoregressive image generation in a pretraining setting, where it improves performance over standard teacher-forcing—highlighting its broader potential across diverse domains and training settings.

## 2   Related Work

**Limitations of Next-Token Prediction**   Bachmann and Nagarajan [2024] introduce a path-finding task on star-graphs to highlight key limitations of standard next-token prediction (i.e., teacher forcing). Their findings reveal that next-token prediction encourages shortcuts, making the underlying task intractable and reducing validation accuracy to random-guessing levels. Interestingly, this "cheating" behavior can be mitigated by multi-token prediction with lookahead embeddings, as in Monea et al. [2023]. While the task is an extreme case, alternative transformer architectures can solve it [Frydenlund, 2024], reinforcing the intuition that tasks requiring planning might need tailored training objectives [Bubeck et al., 2023].

**Multi-token and Lookahead Prediction** Several works have explored decoding multiple future tokens, primarily to accelerate inference rather than improve generation quality [Stern et al., 2018, Monea et al., 2023, Li et al., 2024, Cai et al., 2024]. In contrast, Qi et al. [2020] propose a multi-token prediction pre-training objective that enhances performance on some sequence-to-sequence tasks. However, their method scales poorly to large decoder models because their multi-stream attention mechanism becomes computationally expensive as prediction depth increases. More recently, Gloeckle et al. [2024] proposed an architecture with multiple parallel decoding heads for multi-token prediction, leading to better generative performance (mostly in coding tasks). Liu et al. [2024] modified this approach to use sequential decoding heads instead. Both studies suggest that multi-token prediction helps by providing richer supervision, better information sharing between tokens, and implicit planning in hidden states. However, these benefits are mainly observed during pretraining, with limited success in fine-tuning scenarios—a gap our work addresses.

A less related line of research trains autoregressive models on permuted sequences [Yang et al., 2019, Pannatier et al., 2024, Kakogeorgiou et al., 2024, Yu et al., 2024, Pang et al., 2024]. By predicting next tokens in shuffled order, these methods force the model to recover distant dependencies in the original sequence. While conceptually interesting, they differ significantly from our approach.

**Trainable Dummy / Register Tokens** Recent work investigates the use of trainable tokens as transformer inputs, revealing emergent properties. Burtsev et al. [2020] prepend dummy tokens to prompts to induce working memory, but observe minimal gains. Goyal et al. [2024] demonstrate that appended trainable tokens can improve performance by increasing operations between tokens, thus encouraging deeper "thinking". Pfau et al. [2024] study the conditions under which a language model can leverage the extra computation provided by the dummy tokens. Related efforts incorporate planning tokens trained to predict latent reasoning steps [Wang et al., 2023, Yin et al., 2024]. In the vision domain, Darcet et al. [2024] employ register tokens during pretraining to prevent the appearance of high-norm artifacts. Similarly to these works, we use learnable tokens—but we also associate them with an auxiliary multi-token prediction objective to create a denser training signal.

More related to our work, Monea et al. [2023] appends lookahead tokens to the input sequence and trains them (while freezing the base model) to enable parallel decoding for speculative sampling. Bachmann and Nagarajan [2024] adapt this idea to path-finding on star-graphs, by simultaneously predicting multiple answer tokens from a prefix. While effective for these specific graph problems, the approach's complete "teacherless" training paradigm and parallel inference requirements make it fundamentally unsuitable for the broader range of generative tasks we consider. Unlike these methods, our approach only uses register tokens during training to enhance supervision, without modifying the inference procedure or requiring additional compute at test time.

## 3 Method

### 3.1 Preliminaries

**Next-Token Prediction** Building on the foundational work of Shannon [1948, 1951], next-token prediction remains the core objective for autoregressive language models. Given a sequence $(x_1, \ldots, x_t)$, the model is trained to predict the next token $x_{t+1}$ by maximizing the joint probability under left-to-right factorization:

$$P(x_1, \ldots, x_T) = \prod_t P(x_{t+1} \mid x_{\leq t}), \tag{1}$$

where $T$ is the sequence length. For a model $P_\theta$ and dataset $\mathcal{D}$, the training objective will be to minimize the the expected negative log-likelihood loss over the dataset:

$$\mathcal{L}_{\text{ntp}} = \mathbb{E}_{\mathcal{D}} \left[ -\sum_t \log P_\theta(x_{t+1} \mid x_{\leq t}) \right]. \tag{2}$$

**Multi-Token Prediction** In contrast, Gloeckle et al. [2024] propose predicting multiple future tokens per position. Their objective minimizes:

$$\mathcal{L}_{\text{mtp}} = \mathbb{E}_{\mathcal{D}} \left[ -\sum_t \log P_\theta(x_{t+1:t+d_{\max}} \mid x_{\leq t}) \right], \tag{3}$$

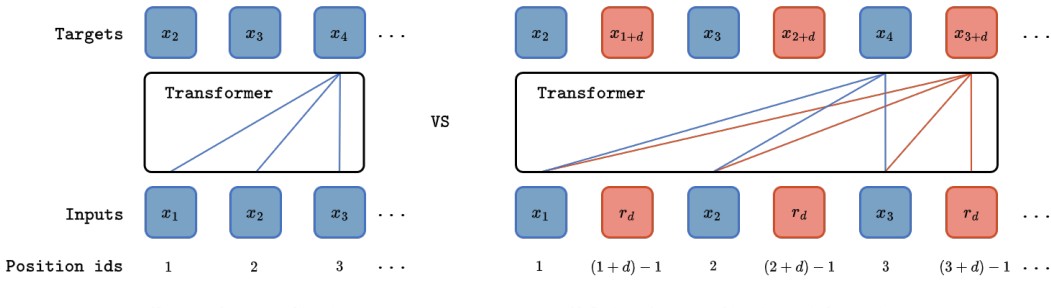

Figure 1: **Next-token prediction vs. Multi-token prediction with registers** (MuToR). The transformer block represents any decoder-only autoregressive model, with colored lines indicating permitted attention connections between tokens. **Left**: Standard next-token prediction, where each $x_t$ predicts $x_{t+1}$ conditioned on preceding tokens. **Right**: MuToR interleaves register tokens $r_d$ to predict tokens $d$ steps ahead ($x_{t+d}$), conditioned only on previous *regular* tokens. Register tokens are assigned position ids (e.g., $t + d - 1$ for $r_d$ targeting $x_{t+d}$) that mimic next-token prediction. Regular tokens follow the standard next-token prediction formulation, unaffected by the registers.

where we denote the maximum prediction horizon as $d_{\max}$, to align notation with our method. Recent implementations utilize additional transformer heads—either parallel [Gloeckle et al., 2024] or sequential [Liu et al., 2024].

## 3.2 Our approach

In this work, we introduce MuToR, an alternative multi-token prediction method (illustrated in Figure 1). Our approach inserts interleaved learnable tokens—termed *registers*, following Darcet et al. [2024]—into training sequences, where each register is tasked with predicting a future token at a uniformly sampled offset $d$. By optimizing this auxiliary prediction objective alongside the primary next-token prediction task, the model benefits from richer supervisory signals, which enhances the quality of learned representations.

**Register Tokens**  Let $x = (x_1, x_2, \ldots, x_T)$ be a training sequence. We augment $x$ by interleaving[1] register tokens $r_d$, yielding:

$$x' = (x_1, r_d, x_2, r_d, \ldots, x_{T-1}, r_d, x_T), \quad (4)$$

where each $r_d$ predicts the future token at offset $d$[2]. By default, all $r_d$ share a single learnable embedding, adding minimal trainable parameters, while the target offset $d$ is specified via the register's position id (detailed later). The augmented sequence $x'$ is processed by the transformer $P_\theta$, which shares all components—embeddings, layers, and prediction head—between the regular tokens $x_t$ and register tokens $r_d$.

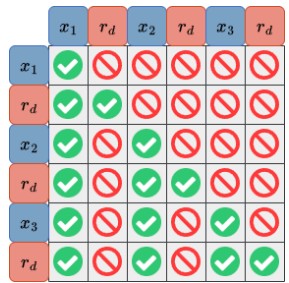

Figure 2: MuToR's attention mask. Each cell indicates whether the row can attend to the column.

**Attention Masking**  We modify the causal attention mask to satisfy two conditions: (i) regular tokens $x_t$ cannot attend to any register tokens, and (ii) register tokens cannot attend to other register tokens in the sequence (see Figure 2). This pattern preserves the standard next-token prediction objective (Equation 2) for regular tokens, as their representations remain unaffected by registers. As a result, the registers can be discarded during inference.

**Multi-token Prediction with Registers**  Each register token $r_d$ inserted after $x_t$ predicts the future token $x_{t+d}$, with offset $d$ sampled uniformly per sequence from $\{1, \ldots, d_{\max}\}$, where $d_{\max}$

---

[1]The interleaving pattern is flexible—e.g., registers may be sparse or appear consecutively.

[2]For simplicity, we use a fixed $d$ per sequence, though registers with different offsets could be mixed.

determines the maximum prediction horizon. The auxiliary register loss over dataset $\mathcal{D}$ is:

$$\mathcal{L}_{\text{reg}} = \mathbb{E}_{\mathcal{D}} \left[ -\sum_t \log P_\theta(x_{t+d} \mid x_{\leq t}, r_d) \right]. \tag{5}$$

Our attention masking ensures predictions depend only on preceding regular tokens, excluding other registers. This approach enables flexible prediction horizons while maintaining parameter efficiency, as all register tokens—regardless of $d$—share a single learnable embedding.

**Position Embeddings for Registers**   Since we do not use specialized heads for future token prediction, we encode prediction offsets through positional bias. We utilize the token indices in the original (register-free) sequence to compute positional embeddings. While regular tokens $x_t$ keep their natural positions $t$, each register $r_d$ inserted after $x_t$ (predicting $x_{t+d}$) receives position $t + d - 1$ (see position ids in Figure 1). This matches the position id of the regular token that would normally predict $x_{t+d}$ under standard next-token prediction.

Our implementation focuses on RoPE [Su et al., 2024], the dominant positional encoding in modern language [Touvron et al., 2023b] and autoregressive image models [Sun et al., 2024]. However, our position manipulation works with any embedding scheme (sinusoidal, relative, etc.), requiring no architectural changes while effectively encoding prediction offsets through positional bias.

**Overall Training Loss**   We jointly optimize the standard next-token prediction loss $\mathcal{L}_{\text{ntp}}$ and the auxiliary register loss $\mathcal{L}_{\text{reg}}$. The overall loss combines these two objectives through a weighted sum:

$$\mathcal{L}_{\text{mtp}} = (1 - a)\mathcal{L}_{\text{ntp}} + a\mathcal{L}_{\text{reg}}, \tag{6}$$

where $a \in (0, 1)$ controls the relative contribution of each loss term.

**Inference**   During inference, we discard register tokens entirely, leaving the model's computational graph and latency unchanged. This is made possible by our attention masking strategy, which prevents regular tokens from ever attending to registers during training. Unlike prior approaches (Goyal et al. [2024], Pfau et al. [2024]) that use inserted tokens to increase inference computation, our method maintains the standard autoregressive process without modification.

## 3.3   Adaptation in Language Modeling

A key application of our method is supervised fine-tuning of pre-trained language models. For generative tasks (e.g., mathematical reasoning), where datasets contain (prefix, answer) pairs—with answer sequences potentially including chain-of-thought tokens—we interleave register tokens only within the answer sequence. This aligns with standard practice where prefix predictions are excluded from loss computation. Beyond this adaptation, the method follows the implementation described in subsection 3.2 without modification.

## 3.4   Adaptation in Autoregressive Image Generation

Autoregressive transformers achieve strong performance on image generation by modeling discrete visual tokens [Esser et al., 2021, Sun et al., 2024]. A VQ-VAE tokenizer [Van Den Oord et al., 2017] first encodes an image into a 2D token grid $x^{\text{2D}} \in \mathbb{Z}^{h \times w}$, which is then flattened into a 1D sequence $x$ via raster-scan ordering. The model then learns autoregressive prediction conditioned on $c$ (either class labels or captions).

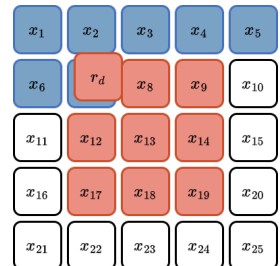

Figure 3: The 2D neighborhood of possible prediction targets (depicted in red) for a register token. The register $r_d$ is inserted after $x_7$, and $d_{\text{max\_2D}}$ is set to 3.

We adapt MuToR to images by modifying the offset sampling to respect the 2D image structure. For each sequence $x$, we sample a 2D offset pair $(d_h, d_w)$, with both $d_h$ and $d_w$ drawn uniformly from $\{1, \ldots, d_{\text{max\_2D}}\}$, excluding $(d_h, d_w) = (1, 1)$, as it denotes the image token after which the register is inserted (see Figure 3). We then compute the rasterized offset as $d = (d_h - 1) \cdot w + d_w - 1$. As in subsection 3.2, each register token $r_d$ predicts the token $d$ steps ahead in the sequence, with all other

components (attention masking, positional embeddings, and loss) implemented identically—except for the additional conditioning on $c$. Each register thus predicts one of $d_{\text{max\_2D}}^2 - 1$ possible future tokens.

This 2D extension enriches the training signal by capturing spatial dependencies inherent in visual data, while requiring minimal architectural changes. Unlike prior multi-token prediction approaches that would require multiple additional prediction heads (one for each possible 2D offset), `MuToR` achieves this capability with negligible parameter overhead.

## 4 Results

### 4.1 Language Modeling

**Experimental Setup**    We focus on mathematical reasoning with chain-of-thought and abstractive summarization, two challenging generative tasks that provide a rigorous testbed for our method. To evaluate performance, we fine-tune two pretrained decoder-only language models: Gemma 2B [Gemma Team et al., 2024] and Llama 3 8B [Grattafiori et al., 2024]. Our experiments target three widely used mathematical reasoning benchmarks: GSM8K [Cobbe et al., 2021], MATH500 [Lightman et al., 2023], and AQUA-RAT [Ling et al., 2017]. For fine-tuning, we use curated subsets from OpenMathInstruct-2 [Toshniwal et al., 2025], a high-quality dataset derived from GSM8K and MATH training samples. Specifically, we filter the 1M and 2M splits to isolate grade school (GSM-style) or MATH-style problems, referring to them as *1M-GSM*, *2M-GSM*, and *1M-MATH*. We also fine-tune on the original GSM8K and AQUA-RAT training splits. Additional details about the experimental setup are provided in Appendix A.1. As for summarization, we target the following benchmarks: SAMSum [Gliwa et al., 2019] and DialogSum [Chen et al., 2021]. For mathematical tasks, we measure exact-match accuracy; for summarization, we use ROUGE scores [Lin, 2004].

**Baselines**    We mainly consider two baselines: (i) `Next-Token`, the standard fine-tuning recipe using the next-token prediction objective, and (ii) `Multi-Token` [Gloeckle et al., 2024], adapted for fine-tuning by adding $d_{\text{max}} - 1$ auxiliary prediction heads and applying a loss-weighting strategy for these heads, similar to our method. To ensure a fair comparison, for both `Multi-Token` and `MuToR`, we tune the number of predicted future tokens, $d_{\text{max}}$, alongside the auxiliary loss coefficient. Implementation details are provided in Appendix A.1.

**Comparative Results**    Tables 1 and 2 present our results, showing only the best configurations for `Multi-Token` and our `MuToR` method. In mathematical reasoning (Table 1), our approach consistently outperforms both baselines, including `Multi-Token`, which introduces a substantial number of additional trainable parameters (see Table 4). Moreover, `MuToR`'s gains are preserved across varying training set sizes, demonstrating its effectiveness even in settings with high-quality fine-tuning data. In contrast, `Multi-Token`'s benefits seem to diminish with larger models or more data. For summarization (Table 2), our `MuToR` method improves all ROUGE scores, achieving superior results over `Multi-Token`, demonstrating broad applicability in sequence-to-sequence generative tasks.

**Comparison with DeepSeek's Sequential Multi-Token Prediction**    We further compare `MuToR` against the multi-token prediction variant [Liu et al., 2024], that uses sequential heads to predict up to $d_{\text{max}}$ tokens ahead while maintaining the causal chain of prediction. The results, presented in Table 16 (Appendix C.1), indicate that `DS-Multi-Token`[3] offers only marginal improvements over the parallel `Multi-Token` variant [Gloeckle et al., 2024], despite its greater architectural complexity. Importantly, `MuToR` outperforms both methods, demonstrating better adaptability on the supervised fine-tuning setting. As both multi-token prediction variants yield similar performance, we stick with `Multi-Token` for most of the experiments and ablations that are included in this paper.

**Matching the Training-Time Compute**    Our method increases sequence length during training (and thus training compute) by inserting register tokens. To ensure gains are not due to higher compute alone, we train the baselines for more epochs (roughly doubling compute), thus matching or even

---

[3]For brevity, we refer to the sequential multi-token prediction as `DS-Multi-Token`, since it was first used when training Deepseek-V3.

Table 1: Downstream accuracy (%) in mathematical reasoning benchmarks. The subheaders refer to the training split used in each experiment. Results for Gemma 2B are averaged over 3 seeded runs. `MuToR` offers a consistent improvement over both standard `Next-Token` and `Multi-Token`.

| Model | Method | GSM8K | | | MATH500 | AQUA-RAT |
|-------|--------|-------|-------|-------|---------|----------|
| | | GSM8K | 1M-GSM | 2M-GSM | 1M-MATH | AQUA-RAT |
| Gemma 2B | `Next-Token` | 38.87 | 66.09 | 69.02 | 26.73 | 40.16 |
| | `Multi-Token` | 40.66 | 66.69 | 69.02 | 26.87 | 38.45 |
| | `MuToR` (ours) | **42.10** | **68.33** | **70.56** | **28.13** | **41.73** |
| Llama3 8B | `Next-Token` | 66.41 | 85.74 | 87.11 | 41.4 | - |
| | `Multi-Token` | 66.56 | 85.67 | 86.35 | 42.6 | - |
| | `MuToR` (ours) | **67.85** | **87.11** | **87.64** | **43.4** | - |

Table 2: Experimental results for fine-tuning Gemma 2B on abstractive summarization benchmarks. We select the checkpoint with the higher ROUGE-L in the validation set, and report ROUGE scores on the test set.

| Method | SAMSum | | | DialogSum | | |
|--------|---------|---------|---------|-----------|---------|---------|
| | ROUGE-1 | ROUGE-2 | ROUGE-L | ROUGE-1 | ROUGE-2 | ROUGE-L |
| `Next-Token` | 51.47 | 27.29 | 43.23 | 47.23 | 20.91 | 38.77 |
| `Multi-Token` | 51.90 | 27.44 | 43.50 | 47.98 | 21.23 | 39.25 |
| `MuToR` (ours) | **52.32** | **28.08** | **44.09** | **48.22** | **21.71** | **39.48** |

exceeding the compute of `MuToR`. As shown in Tables 12 and 13 (Appendix B), extending training does not yield further improvements for the baselines and, in some instances, can even degrade performance due to mild overfitting. These results suggest that integrating `MuToR` into the fine-tuning pipeline offers a more effective approach to leveraging increased training compute, when the available data is fixed—a common real-world constraint in many academic and industrial applications.

**Matching Compute with Additional Data**   Through our experiments, we focus on the data-constrained setting both for practical reasons (limited resources) and methodological clarity (ensuring a fair comparison with baselines). However, we also investigate the possibility of matching `MuToR`'s increased training-time compute by leveraging additional data for the `Next-Token` baseline. The result (presented in Table 3), shows that `MuToR` *can surpass gains from additional data*. While a thorough exploration of the data-abundant regime is left for future work, this preliminary finding highlights `MuToR`'s potential beyond data-constrained scenarios. More details are provided in the Appendix B.1.

Table 3: Downstream accuracy (%) on GSM8K. 1M-GSM-extra split corresponds to matching `MuToR`'s training compute by utilizing additional fine-tuning data.

| Method & Data split | GSM8K |
|---------------------|-------|
| `Next-Token (1M-GSM)` | 66.09 |
| `Next-Token (1M-GSM-extra)` | 66.69 |
| `MuToR (1M-GSM)` | **68.33** |

**Integration in Parameter-Efficient Fine-tuning**   We test our method with LoRA [Hu et al., 2022]: for both `Next-Token` and our `MuToR` method, we apply rank-32 adapters to all linear layers (approximately 39M trainable parameters for Gemma 2B; register tokens in `MuToR` add a negligible number of parameters). As shown in Table 5, our `LoRA-MuToR` approach improves accuracy over standard LoRA fine-tuning across both training splits. Interestingly, `LoRA-MuToR` *matches or even exceeds the full fine-tuning* `Next-Token` performance, demonstrating its utility in PEFT setups. In contrast, the `Multi-Token` approach is less compatible with PEFT settings, as it requires training several additional transformer layers from scratch.

**Impact of Maximum Lookahead Offset**   The key hyperparameter $d_{\max}$ in our `MuToR` method controls how many tokens ahead the registers predict. Larger values offer richer supervision but increase task difficulty. Experiments (Table 4) show that $d_{\max} = 4$ is optimal for this particular setting (in general the optimal value may depend on the training data and the downstream task). Notably, as $d_{\max}$ increases, `Multi-Token`'s performance degrades, barely beating `Next-Token`

Table 4: Downstream accuracy (%) with respect to the maximum offset $d_{\max}$, using Gemma 2B. #Add. Param. denotes the additional trainable parameters for `Multi-Token` and `MuToR` (in approximation).

| $d_{\max}$ | #Add. Param. | Method | GSM8K | |
|---|---|---|---|---|
| | | | GSM8K | 1M-GSM |
| 1 | - | `Next-Token` | 38.87 | 66.09 |
| 2 | 110M | `Multi-Token` | 40.66 | 66.69 |
| | 2K | `MuToR` (ours) | 41.93 | 67.15 |
| 3 | 220M | `Multi-Token` | 40.59 | 66.36 |
| | 2K | `MuToR` (ours) | 41.60 | 68.01 |
| 4 | 330M | `Multi-Token` | 39.78 | 65.53 |
| | 2K | `MuToR` (ours) | **42.10** | **68.33** |
| 6 | 550M | `Multi-Token` | 40.23 | 65.55 |
| | 2K | `MuToR` (ours) | 39.90 | 68.16 |

Table 5: Downstream accuracy (%) in a PEFT scenario, using Gemma 2B and LoRA.

| Method | GSM8K | |
|---|---|---|
| | GSM8K | 1M-GSM |
| Full fine-tuning `Next-Token` | 38.87 | 66.09 |
| `LoRA-Next-Token` | 36.34 | 66.11 |
| `LoRA-MuToR` (ours) | **38.59** | **68.11** |

Table 6: Ablation regarding the register embeddings, using Gemma 2B and $d_{\max} = 4$.

| Register embedding | GSM8K | |
|---|---|---|
| | GSM8K | 1M-GSM |
| Same | **42.10** | **68.33** |
| Different | 41.85 | 68.18 |

when using the *1M-GSM* training split. In comparison, `MuToR`'s gains are maintained across $d_{\max}$ values and training split sizes.

**Shared vs. Different Register Embeddings Per Offset**  We test whether having different register embeddings per offset improves performance. As seen in Table 6, shared embeddings (*Same*), which is the default, perform slightly better than distinct ones (*Different*), suggesting our positional encoding scheme provides sufficient offset information.

## 4.2 Autoregressive Image Generation

**Experimental Setup**  We train LlamaGen-B (111M parameters; Sun et al. 2024) on ImageNet [Deng et al., 2009] at 256×256 resolution, using ADM's preprocessing pipeline [Dhariwal and Nichol, 2021]. The dataset is pre-tokenized using a VQ-VAE tokenizer from Sun et al. [2024].

We compare three approaches: (1) `Next-Token`, a standard next-token prediction baseline; (2) `MuToR-1D` with 1D offsets (as in language modeling); and (3) `MuToR-2D` with 2D offsets (described in subsection 3.4). Implementation details are included in Appendix A.2.

For evaluation, we generate 50,000 images using classifier-free guidance (scale=2.0) [Ho and Salimans, 2022] and compute FID [Heusel et al., 2017], IS [Salimans et al., 2016], Precision, and Recall [Kynkäänniemi et al., 2019] using TensorFlow scripts from Dhariwal and Nichol [2021].

**Results**  Table 7 shows that both `MuToR` variants consistently outperform `Next-Token` in FID and IS across different training iterations. Notably, when comparing under similar training-time compute, the `MuToR-2D` variant at 100K steps surpasses the `Next-Token` model at 200K steps, demonstrating both improved performance and faster convergence.

**Extending the offset to 2D**  The 2D extension in `MuToR-2D` significantly improves performance by leveraging spatial dependencies in the image data (Table 7), despite requiring prediction of much more possible future tokens. This demonstrates that the 2D formulation effectively captures structural patterns, providing richer training signal.

**Scaling down the number of registers during pretraining**  To reduce computational costs while maintaining performance, we investigate using fewer register tokens in `MuToR-2D`. Specifically, we test inserting only 80 randomly placed registers per image, increasing the sequence length by just 30%. As shown in Table 8, this sparse version achieves very similar performance to the full setup (with 256 registers) while requiring less computation. These results demonstrate that substantial performance gains can be obtained with relatively few register tokens and only a modest increase in training compute. They also highlight untapped potential in `MuToR`'s design, particularly regarding optimal register placement and sparsity strategies, which merit further investigation.

Table 7: Conditional generation performance on Imagenet $256 \times 256$ (cfg scale = 2.0). Both $d_{\max}$ and $d_{\max\_2D}$ are set to 4, for `MuToR-1D` and `MuToR-2D` respectively.

| # Iter. | Method | FID ↓ | IS ↑ | Pre. ↑ | Rec. ↑ |
|---|---|---|---|---|---|
| 100K | Next-Token | 7.71 | 146.5 | 0.830 | 0.439 |
|  | MuToR-1D | 7.01 | 155.6 | 0.828 | 0.438 |
|  | MuToR-2D | **6.57** | **163.2** | **0.831** | **0.445** |
| 200K | Next-Token | 6.83 | 158.4 | 0.833 | 0.443 |
|  | MuToR-1D | 6.43 | 163.0 | 0.836 | 0.444 |
|  | MuToR-2D | **5.65** | **183.5** | **0.842** | **0.448** |
| 360K | Next-Token | 6.18 | 171.5 | **0.841** | 0.443 |
|  | MuToR-1D | 5.79 | 178.3 | **0.841** | 0.441 |
|  | MuToR-2D | **5.09** | **195.3** | 0.839 | **0.457** |

**Maximum Offset Analysis**  We examine how expanding the prediction neighborhood in `MuToR-2D` affects performance by varying $d_{\max\_2D}$, which determines how many future tokens each register must predict. Using 80 randomly placed registers per sequence, we test different $d_{\max\_2D}$ values. Table 9 shows that $d_{\max\_2D} = 4$ achieves optimal performance, while increasing it leads to degrading results. In this optimal setup, each register predicts up to 15 future tokens—a prediction horizon that would require 14 additional transformer heads in prior multi-token approaches, making them computationally impractical. This demonstrates `MuToR-2D`'s unique ability to effectively leverage long-range predictions while maintaining training efficiency.

Table 8: Ablation using `MuToR-2D`, $d_{\max\_2D} = 4$, and varying number of registers (# Reg.) inserted in random positions.

| # Iter. | # Reg. | FID ↓ | IS ↑ |
|---|---|---|---|
| 100K | 80 | 6.87 | 162.1 |
|  | 256 | **6.57** | **163.2** |
| 200K | 80 | 5.89 | 175.6 |
|  | 256 | **5.65** | **183.5** |
| 360K | 80 | **5.03** | **195.6** |
|  | 256 | 5.09 | 195.3 |

Table 9: Ablation using `MuToR-2D`, with varying $d_{\max\_2D}$ and 80 registers inserted in random positions. # Targets denote the number of future tokens ($d^2_{\max\_2D} - 1$) that the registers predict during training (see Figure 3).

| # Iter. | $d_{\max\_2D}$ | # Targets | FID ↓ | IS ↑ |
|---|---|---|---|---|
| 100K | 3 | 8 | 7.20 | 150.8 |
|  | 4 | 15 | **6.87** | **162.1** |
|  | 5 | 24 | 7.08 | 154.8 |
| 200K | 3 | 8 | 6.19 | 169.0 |
|  | 4 | 15 | **5.89** | **175.6** |
|  | 5 | 24 | 6.21 | 166.2 |
| 360K | 3 | 8 | 5.56 | 178.3 |
|  | 4 | 15 | **5.03** | **195.6** |
|  | 5 | 24 | 5.53 | 179.2 |

## 4.3 Synthetic Data

We further evaluate `MuToR` on the star-graph path-finding problem [Bachmann and Nagarajan, 2024], which highlights limitations of the next-token prediction objective. Consider a directed graph $G = (n, l)$, where $n$ denotes the number of paths emanating from the start node, $u_{\text{start}}$, and $l$ denotes their length. Given an end node, $u_{\text{end}}$, the model must identify the unique path from $u_{\text{start}}$ to $u_{\text{end}}$.

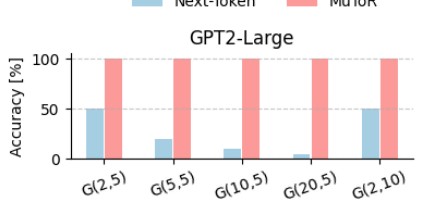

Figure 4: Solve rate (%) of finetuned GPT2-L model on different star graph configurations.

Despite the task's simplicity, transformer models trained by teacher forcing fail to solve it, due to shortcut learning and the loss of meaningful training signal. We thus investigate whether the look-ahead induced by `MuToR`'s objective can provide the necessary supervision to learn the underlying task. To this end, we experiment with fine-tuning a pretrained GPT2 model [Radford et al., 2019] following the setup from Bachmann and Nagarajan [2024]. The relevant implementation details are provided in the Appendix A.3.

**Results** The results are presented in Figure 4. `MuToR` solves the task across various graph configurations, effectively overcoming the "cheating phenomenon" that causes standard teacher forcing to fail. These findings indicate that `MuToR` can be particularly effective in scenarios where some form of shortcut learning applies, recovering valuable training signal.

## 5 Conclusion

We introduced `MuToR`, a simple yet effective approach to multi-token prediction that leverages interleaved, trainable register tokens to predict future targets. `MuToR` enables scalable prediction horizons with minimal parameter overhead and is fully compatible with existing pretrained model architectures, allowing seamless integration into standard fine-tuning pipelines. Empirical results demonstrate that `MuToR` consistently improves performance in both language modeling tasks—such as mathematical reasoning and summarization—and autoregressive image generation, highlighting its versatility across modalities. This positions `MuToR` as a promising foundation for using token-based lookahead mechanisms to propagate richer supervisory signals during training.

**Limitations** It is worth noting that `MuToR` currently uses uniformly interleaved or randomly positioned register tokens—strategies that may not align optimally with the structure or semantics of specific tasks. While this simple placement scheme has proven effective across modalities, it leaves room for substantial improvement. By learning or adapting the placement of register tokens—potentially guided by model uncertainty or task-specific priors—`MuToR` could deliver more targeted supervision with fewer auxiliary tokens, further enhancing efficiency and performance.

## Acknowledgments and Disclosure of Funding

This work has been partially supported by project MIS 5154714 of the National Recovery and Resilience Plan Greece 2.0 funded by the European Union under the NextGenerationEU Program. Hardware resources were granted with the support of GRNET.

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

# A   Implementation Details

In this section, we provide the necessary training details, as well as the hyperparameters used in our experiments.

## A.1   Language Modeling

### A.1.1   Mathematical Reasoning

**Datasets**   GSM8K comprises approximately 8.7K grade-school level math problems, with around 1.3K of them forming the test set. On the other hand, MATH500 test set consists of 500 problems, uniformly sampled from the original MATH test set [Hendrycks et al., 2021], covering more advanced and diverse mathematical domains. Finally, AQUA-RAT includes multiple-choice math problems with chain-of-thought solutions, split among the training set ( $\sim$97K samples), the validation ($\sim$250 samples) and the test set ($\sim$250 samples).

As mentioned in subsection 4.1, we finetune our base models on standard downstream training datasets, such as GSM8K and AQUA-RAT, as well as on curated subsets derived from OpenMathInstruct-2. The utilized training splits are listed below, including information about the employed filtering and the amount of samples:

- GSM8K training set ($\sim$ 7.4K samples),

- 1M-GSM split ($\sim$152K samples), obtained by filtering OpenMathInstruct-2's 1M split for grade school math-like problems,

- 2M-GSM split ($\sim$277K samples), similarly derived from OpenMathInstruct-2's 2M split using the same grade-school filtering criteria,

- 1M-MATH split ($\sim$200K samples), constructed by filtering OpenMathInstruct-2's 1M split for MATH-style problems and randomly sampling 200K of them,

- AQUA-RAT training set ($\sim$97K samples).

Our filtering depends on the source dataset of each sample, which is included in OpenMathInstruct-2's metadata.

**Training details**   Throughout all experiments and for all methods, we use bidirectional attention among the prefix tokens, as proposed in previous works [Dong et al., 2019, Raffel et al., 2020, Kopiczko et al., 2024], since it benefits prefix-answer tasks. We finetune all models for 5 epochs, using AdamW optimizer [Loshchilov and Hutter, 2017] without weight decay and a batch size of 10. We also employ a learning rate scheduler with linear decay and warmup, setting the peak learning rate to be 5e-5 for Gemma 2B and 2e-5 for Llama 3 8B. Both language models are loaded and finetuned in bfloat16 precision. To match our available resources, we filter out training sequences that are longer than 512 tokens[4]. Moreover, all experiments with the 2B model are conducted using three random seeds to ensure statistical robustness. This setting is not replicated for the 8B model though, due to the substantial computational resources that are required.

All experiments with Gemma 2B model are run using a single A100 GPU and gradient accumulation. For the experiments with the Llama 3 8B model, we utilize $5 \times$ A100 and Fully Sharded Data Parallelism (FSDP).

**Evaluation**   During evaluation, we assess performance by using greedy decoding and applying exact match techniques to verify the correctness of the generated answer. For experiments on GSM8K and MATH500, the best-performing checkpoint from each run is then used for comparisons. For experiments on AQUA-RAT, we choose the checkpoint with the highest accuracy on the provided validation set. Since register tokens are not used during inference, the inference process is identical across both our method and the baseline approaches.

---

[4]In 1M-MATH, we keep sequences up to 768 tokens.

### A.1.2 Abstractive summarization

**Datasets** In our experiments, we target SAMSum and DialogSum, two widely used dialogue summarization benchmarks. SAMSum consists of approximately 16K messenger-like conversations with summaries. They are split between the training set ($\sim$14K samples), the validation set ($\sim$818 samples) and the test set ($\sim$819 samples). On the other hand, DialogSum contains approximately 14K conversation-summary pairs, focusing more on daily-life formal conversations, such as business negotiations. These pairs are split between the training set ($\sim$12.5K samples), the validation set ($\sim$500 samples) and the test set ($\sim$1.5K samples).

**Training details** Across all experiments, we finetune the model for 3 training epochs, following the same setup (optimization details, weights' precision), with the mathematical reasoning experiments. We filter out training sequences that are longer than 768 tokens. Since these experiments are conducted with the 2B model, we utilize a single A100 GPU and gradient accumulation.

**Evaluation** We calculate ROUGE scores against the ground truth reference summaries, select the best checkpoint with respect to ROUGE-L on the validation set, and report ROUGE-1, ROUGE-2 and ROUGE-L scores on the test set.

### A.1.3 Best performing configurations

In Tables 10 and 11, we provide the best performing configurations ($d_{\max}$ and $a$) for `MuToR`, across all the training splits that were utilized in our experiments. For `Multi-Token`, after carefully tuning the same hyperparameters, we found that the optimal configuration across all experiments is $d_{\max} = 2, a = 0.1$.

Table 10: `MuToR`'s best performing hyperparameters for Gemma 2B model.

| Training data | $d_{\max}$ | $a$ |
| --- | --- | --- |
| GSM8K | 4 | 0.3 |
| 1M-GSM | 4 | 0.3 |
| 2M-GSM | 6 | 0.1 |
| 1M-MATH | 4 | 0.3 |
| AQUA-RAT | 4 | 0.3 |
| SAMSum | 3 | 0.5 |
| DialogSum | 4 | 0.5 |

Table 11: `MuToR`'s best performing hyperparameters for Llama 3 8B model.

| Training data | $d_{\max}$ | $a$ |
| --- | --- | --- |
| GSM8K | 4 | 0.3 |
| 1M-GSM | 4 | 0.3 |
| 2M-GSM | 6 | 0.1 |
| 1M-MATH | 4 | 0.3 |

### A.2 Autoregressive Image Generation

**Training details** In the autoregressive image generation experiments, we pretrain LlamaGen-B model (approximately 111M parameters). Specifically, the model itself is an autoregressive decoder-only transformer, similar to Llama [Touvron et al., 2023a]. It utilizes RoPE as positional bias, extended to two dimensions. The model also involves learnable class embeddings, which are used as conditionals for image generation. During training, the class embeddings are dropped with a probability of 0.1, to enable the use of classifier-free guidance at inference time.

To tokenize the image patches, we use a VQ-VAE tokenizer provided by Sun et al. [2024], with codebook size equal to 16384 and embedding dimension equal to 8. Unlike LlamaGen, we pre-tokenize the dataset with ADM's preprocessing scheme [Dhariwal and Nichol, 2021], resulting in two crops per image.

Both `Next-Token` baseline and `MuToR` are trained for 360K update steps, using AdamW optimizer with $\beta_1 = 0.9, \beta_2 = 0.95$ and weight decay $= 0.05$. We employ a constant learning rate, equal to 0.0004, and a batch size of 1024. All experiments are run using $8 \times$ H100 GPUs and the Distributed Data Parallel (DDP) framework.

For `MuToR`'s implementation, we empirically tune the loss coefficient $a$ to be 0.5, so that $\mathcal{L}_{\mathrm{reg}}$ has equal contribution with $\mathcal{L}_{\mathrm{ntp}}$ ( Equation 6). This indicates that the auxiliary loss provides valuable

supervision that the model leverages during pretraining, to improve its learned representations. In the experiments that fewer registers are inserted, we sample random positions for each training sample.

**Evaluation**   To benchmark generative performance, we sample 50,000 images at $256 \times 256$ resolution, setting temperature $= 1.0$ and a fixed random seed for fair comparison. We use classifier-free guidance [Ho and Salimans, 2022] with scale $s = 2.0$, which was reported as an optimal value in the original LlamaGen. Thus, the logit $l_g$ is formed as such: $l_g = l_u + 2(l_c - l_u)$, where $l_u$ denotes the unconditional logit (with the class embedding dropped) and $l_c$ denotes the conditional logit. Then, the generated samples are used to calculate the performance metrics, using the Tensorflow scripts from Dhariwal and Nichol [2021].

### A.3  Synthetic Data

We setup our experiments using the official implementation from Bachmann and Nagarajan [2024] (regarding the model's architecture and optimization). All model configurations (`Next-Token` and `MuToR`) are trained for a sufficient number of epochs, using a single A100 GPU.

`MuToR`'s implementation for the star graphs problem follows subsection 3.3, with a task-specific modification; we sample the offset $d$ from $\{2, \ldots, d_{\max}\}$, thus excluding the next-token from the register's prediction. In this way, we prevent the registers from learning the "Clever Hans Cheat", and enable them to focus on *planning* look-ahead predictions. In these experiments, we set $d_{\max} = 4$ and $a = 0.5$ for the graphs with path length $= 5$ and $d_{\max} = 6$, $a = 0.3$ for graphs with path length $= 10$, which are more challenging.

## B   Matching the Training Compute

As mentioned earlier, `MuToR` inserts additional tokens in the training sequence, thereby increasing its length and the overall training compute. To estimate the total training-time compute, we follow the widely used approximation:

$$C \approx 6 \times N \times D, \tag{7}$$

where $C$ denotes the compute (in FLOPs), $N$ denotes the model's parameters, and $D$ denotes the total number of tokens processed during training. This formulation is supported by both empirical studies and theoretical analyses in prior scaling literature [Kaplan et al., 2020, Hoffmann et al., 2022].

While the self-attention mechanism has a theoretical $O(N^2)$ complexity, in practice the compute cost is dominated by the feedforward layers (unless the sequence length $L$ becomes disproportionally larger than the embedding dimension), making the total training compute effectively linear in sequence length. This approximation holds for the Large Language Models and the sequence lengths used in our experiments, since the *tipping points*[5] for these particular models lie far beyond their supported context length.

We further validate this approximation empirically by measuring training wall-clock time. In our 1M-GSM experiments, `MuToR` requires $1.4\times$ the wall-clock time of the `Next-Token` baseline for the same number of epochs. This aligns with expectations, especially since `MuToR` only interleaves register tokens into the answer portion, not the prefix, so the effective sequence length increase is less than $2\times$. As a result, doubling the number of epochs for the baselines leads to equal or higher total compute in the data-constrained setting that we explore.

### B.1   Language Modeling

Tables 12, 13 and 14 report the results of fine-tuning baseline methods for twice the number of training epochs than `MuToR`, in order to investigate whether improved downstream performance can be derived from increasing the training compute. Due to our limited computational resources, we keep the data fixed in these experiments, thus simulating a data-constrained setting.

**Matching the Training Compute via Additional Data**   In Table 3 (subsection 4.1), we report results where the baselines are trained with additional data to match `MuToR`'s training compute.

---

[5]As *tipping points* we refer to the sequence lengths at which attention cost equals the feedforward cost.

Table 12: Gemma 2B: downstream accuracy (%) in mathematical reasoning benchmarks across different number of training epochs. The subheaders refer to the training split used in each experiment. Results are averaged across three seeded runs.

| Method | # Epochs | GSM8K | | MATH500 |
| --- | --- | --- | --- | --- |
| | | GSM8K | 1M-GSM | 1M-MATH |
| Next-Token | 5 | 38.87 | 66.09 | 26.73 |
| | 10 | 37.91 | 65.07 | 27.07 |
| Multi-Token | 5 | 40.66 | 66.69 | 26.87 |
| | 10 | 39.98 | 64.92 | 26.73 |
| MuToR (ours) | 5 | **42.10** | **68.33** | **28.13** |

Table 13: Gemma 2B: ROUGE metrics comparison across different number of training epochs. We select the checkpoint with the higher ROUGE-L in the validation set, and report ROUGE scores on the test set.

| Dataset | Method | #Epochs | ROUGE-1 | ROUGE-2 | ROUGE-L |
| --- | --- | --- | --- | --- | --- |
| SAMSum | Next-Token | 3 | 51.47 | 27.29 | 43.23 |
| | | 5 | 51.67 | 27.64 | 43.32 |
| | Multi-Token | 3 | 51.90 | 27.44 | 43.50 |
| | | 5 | 51.04 | 26.35 | 42.48 |
| | MuToR (ours) | 3 | **52.32** | **28.08** | **44.09** |
| DialogSum | Next-Token | 3 | 47.23 | 20.91 | 38.77 |
| | | 5 | 46.92 | 20.39 | 38.25 |
| | Multi-Token | 3 | 47.98 | 21.23 | 39.25 |
| | | 5 | 47.46 | 20.76 | 38.97 |
| | MuToR (ours) | 3 | **48.22** | **21.71** | **39.48** |

Table 14: Llama 3 8B: downstream accuracy (%) in mathematical reasoning benchmarks across different number of training epochs. The subheaders refer to the training split used in each experiment.

| Method | # Epochs | GSM8K | |
| --- | --- | --- | --- |
| | | GSM8K | 1M-GSM |
| Next-Token | 5 | 66.41 | 85.74 |
| | 10 | 64.74 | 85.74 |
| Multi-Token | 5 | 66.56 | 85.67 |
| | 10 | 65.20 | 84.98 |
| MuToR (ours) | 5 | **67.85** | **87.05** |

Specifically, we construct 1M-GSM-extra split by augmenting 1M-GSM with samples from 2M-GSM, resulting in approximately $1.4\times$ more training samples—consistent with MuToR's $1.4\times$ higher training time. The reported accuracies are obtained with Gemma 2B and are averaged over three seeded runs. All other training details follow subsection A.1.

## B.2 Autoregressive Image Generation

As discussed in subsection 4.2, MuToR-2D demonstrates substantial performance gains even when using only 80 register tokens. In this configuration, training compute increases by approximately 30%. To ensure that our performance gains cannot be matched by increasing the pretraining iterations, we also train the baseline Next-Token for 500K steps—corresponding to a similar increase in

computational cost. As shown in Table 15, `MuToR-2D` delivers significantly better generation quality under comparable compute constraints.

Table 15: Conditional generation performance with respect to training iterations. We compare against the `MuToR-2D` configuration using 80 randomly placed registers, which results in approximately a 30% increase in training compute.

| Method | # Iter. | FID ↓ | IS ↑ |
|---|---|---|---|
| Next-Token | 360K | 6.18 | 171.5 |
| | 500K | 5.82 | 175.8 |
| MuToR-2D (ours) | 360K | **5.03** | **196.6** |

# C    Additional Experiments

## C.1    Comparison with Sequential Multi-Token Prediction

We adapt the sequential multi-token prediction method (`DS-Multi-Token`) to the fine-tuning task by employing $d_{max} - 1$ additional transformer layers. Our implementation closely follows Liu et al. [2024], and all training hyperparameters are identical with those described in subsection A.1. We fine-tune Gemma 2B with `DS-Multi-Token` on mathematical reasoning datasets, including GSM8K, 1M-GSM and 1M-MATH. The hyperparameters $d_{max}$ and $a$ are empirically tuned to 2 and 0.1 respectively.

Table 16 reports results averaged over three seeded runs. Notably, `DS-Multi-Token` performs on par with the parallel `Multi-Token` baseline [Gloeckle et al., 2024], while requiring even more additional trainable parameters. `MuToR` consistently outperforms both methods, using only negligible extra parameters, thus suggesting its superiority when integrated in the supervised fine-tuning stage.

Table 16:  Downstream accuracy (%) in mathematical reasoning benchmarks, using Gemma-2B. The subheaders refer to the training split used in each experiment. All results are averaged over 3 seeded runs.

| Method | # Add. Params. | GSM8K | | MATH500 |
|---|---|---|---|---|
| | | GSM8K | 1M-GSM | 1M-MATH |
| Next-Token | - | 38.87 | 66.09 | 26.73 |
| Multi-Token | 110M | 40.66 | 66.69 | 26.87 |
| DS-Multi-Token | 118M | 40.61 | 66.97 | 27.00 |
| MuToR (ours) | 2K | **42.10** | **68.33** | **28.13** |

Table 17:  Effect of varying the loss coefficient $a$, while fixing $d_{max} = 4$. The results are obtained from single run experiments.

| $a$ | GSM8K | |
|---|---|---|
| | GSM8K | 1M-GSM |
| 0.5 | 40.78 | 67.39 |
| 0.4 | 40.94 | 68.00 |
| 0.3 | **42.00** | **68.38** |
| 0.2 | 41.24 | 67.55 |
| 0.1 | 40.33 | 67.32 |
| 0 (Next-Token) | 38.43 | 66.33 |

## C.2 Impact of the Loss Coefficient

We analyze the effect of the auxiliary loss coefficient $a$, which controls the relative weight between $\mathcal{L}_{\mathrm{ntp}}$ and $\mathcal{L}_{\mathrm{reg}}$. In these experiments, we fine-tune Gemma 2B on mathematical reasoning datasets, while fixing $d_{\max} = 4$. As shown in Table 17, setting $a = 0.3$ yields optimal downstream performance, with both higher and lower values leading in mild degradation.

This trend is expected, as an optimal balance should exist between next-token and multi-token prediction supervisory signals. However, the optimal value for $a$ likely depends on factors like the model, task complexity, data distribution, and $d_{\max}$. For example, in autoregressive image generation, setting $a = 0.5$ achieves better generation quality, likely because the multi-token prediction objective captures rich spatial dependencies.

# D   Broader Impact

Improving large language models can enhance their accuracy, efficiency, and safety, enabling more reliable applications across education, healthcare, and communication. However, it is important to remain mindful of potential risks such as misuse or bias amplification and to continue developing responsible deployment practices.

