# OpenReview forum: "Multi-Token Prediction Needs Registers"
_NeurIPS.cc/2025/Conference — NeurIPS 2025 poster_

### Official Review · Reviewer_sXdq · 2025-06-29

**Clarity:** 4
**Significance:** 3
**Originality:** 3
**Rating:** 4
**Confidence:** 5

**Summary:**

The paper introduces multi-token prediction with registers (MuToR). The method offers advantages including (1) no architectural changes (2) fine-tuning compatibility and (3) scalable prediction horizons. The authors validate MuToR across various settings, including Supervised Fine-Tuning (SFT), Parameter-Efficient Fine-Tuning (PEFT), and image auto-regressive generation. Additionally, ablation studies are conducted to analyze the influence of different components, such as the use of same versus different registers and the maximum prediction distance $d_{max}$.

**Questions:**

1. How does the method compare to other types of MTP like sequential MTP from DeepSeek V3?
2. What is the influence of the ratio of $L_{ntp}$ and $L_{reg}$?

**Ethical Concerns:**

["NO or VERY MINOR ethics concerns only"]

**Final Justification:**

The rebuttal addressed my concerns. Therefore, I maintain my rating as Borderline accept.

**Limitations:**

yes

**Quality:**

3

**Strengths And Weaknesses:**

# Strengths
1. MuToR is a simple yet effective method, demonstrating better performance over both parallel MTP and NTP across diverse settings, including SFT, PEFT, and auto-regressive image generation.
2. The ablation experiments are comprehensive, effectively analyzing the influence of individual components and yielding notable observations, such as the finding that using identical registers is more effective than using different ones.
3. The paper is clearly and neatly written.

# Weakness
1. The baselines seems to be limited. Why not compared to the sequential MTP from DeepSeek V3? I think the method is also mentioned in related works.

---

> ### Author Rebuttal · Authors · 2025-07-31
>
> We thank the reviewer for their positive and constructive feedback.   We are pleased that they found our method to be "simple yet effective" and that they appreciated our ablation analysis and writing style. Below we address the concerns they raised.
>
> ---
> ## **Question 1**
> > How does the method compare to other types of MTP like sequential MTP from DeepSeek V3?
>
> We did not include a comparison with DeepSeek-V3’s sequential MTP[1] in the original submission due to the complexity of adapting its sequential-head architecture for finetuning pretrained LLMs. Following the reviewer’s suggestion, we have now implemented and evaluated this baseline.
> Using the Gemma 2B model, we conducted experiments on the GSM8K training set, the 1M-GSM split and the 1M-MATH split, tuning the key hyperparameters $d_{max}=2$ (maximum prediction depth) and $a=0.1$ (auxiliary loss coefficient).
>
> The results can be seen in the table below; we report average test accuracy (%) over 3 seeded runs, and include the other baselines as well. The headers refer to the finetuning split used.
>
> | Method                         |# Add. Params.| GSM8K       |  1M-GSM     | 1M-MATH |
> |---                             |---           | ---         |  ---        | ---     |
> | *Next-Token*                   |-             |  38.87      |  66.09      | 26.73   |
> | *Multi-Token (Gloeckle et al. [2])* |110M         |  40.66     |  66.69       | 26.87   |
> | Multi-Token (DeepSeek-V3 [1])       |118M          |  40.61      |  66.97      | 27.0    |
> | **MuToR (ours)**               |2K            | **42.10**   |  **68.33**  |**28.13**|
>
>
> As seen in the results, the sequential MTP achieves similar accuracies with the parallel Multi-Token method, but at the cost of greater complexity. Notably, our MuToR method outperforms both of them, suggesting that it is better suited for the downstream finetuning of pretrained Language Models.  In addition, it does so by relying on negligible trainable parameters when comparing to the multi-token baselines(see column 2).
>
> Besides that, the increased complexity of the sequential MTP method (especially as $d_{max}$ scales) limits its applicability, while our method can seamlessly integrate within models without requiring architectural changes. This is particularly significant in domains such as autoregressive image generation, where scaling $d_{max}$ (thus predicting further into the future) would require lots of additional transformer heads for the existing methods [1],[2].
>
> We appreciate the reviewer's feedback and will include these experiments in the final version of the paper.
>
> ---
> ## **Question 2**
> > What is the influence of the ratio of $L_{ntp}$ and $L_{reg}$?
>
> We analyzed the impact of the auxiliary loss coefficient $a$ (controlling the relative weight between $L_{ntp}$ and $L_{reg}$), while fixing $d_{max}=4$.
> We finetuned the Gemma 2B model on GSM8K training set and the 1M-GSM split.
>
> The results are presented in the table below. We report test set accuracy (%) over *1 seeded run* (due to the substantial computational resources needed to replicate each experiment 3 times).
>
> | a         | GSM8K       |  1M-GSM     |
> |---        | ---         |  ---        |
> | 0.5       | 40.78       | 67.39       |
> | 0.4       | 40.94       | 68.00       |
> | **0.3**   |  **42.0**   | **68.38**   |
> | 0.2       | 41.24       | 67.55       |
> | 0.1       |  40.33      | 67.32       |
> | *0 (Next-Token)*| *38.87* | *66.09*   |
>
> We observe that optimal downstream performance is achieved when $a=0.3$, with both higher and lower values leading to mild degradation. These results are expected; an optimal balance should exist between next-token and multi-token prediction supervisory signals.
>
> Notably, the optimal $a$ depends on factors like the pretrained model, task complexity, data distribution, and $d_{max}$. For example, in autoregressive image generation, we found that $a=0.5$ achieves better generation quality, likely because the multi-token prediction objective captures rich spatial dependencies.
> We will include these ablation experiments and insights in the final version of the paper.
>
> We appreciate the reviewer's time and hope we have addressed their concerns.
>
>
> ---
> [1] Liu et al., "DeepSeek-V3 Technical Report", arXiv preprint 2024.
>
> [2] Gloeckle et al., "Better & Faster Large Language Models via Multi-token Prediction." ICML 2024.

---

> > ### Comment · Reviewer_sXdq · 2025-08-09
> >
> > Thanks for providing the experiments. It has addressed my concerns. Therefore, I will keep my rating as Borderline accept.

---

### Official Review · Reviewer_nf2t · 2025-06-30

**Clarity:** 3
**Significance:** 2
**Originality:** 3
**Rating:** 4
**Confidence:** 3

**Summary:**

The paper introduces MuToR (Multi-Token Prediction with Registers), a method that enhances autoregressive models by inserting learnable register tokens during training. Each register token is added in between of the two normal tokens, and predicts a future token d steps ahead. The proposed approach show advantage in both language model fine-tuning and autoregressive image generation.

**Questions:**

1. The proposed method is interesting yet a lot of operations are un-explanable to me. For example, why do you want to insert registers after each token and use it to predict the future token after d steps? If I insert register for maybe every two normal tokens, what would be the difference in performance? Also it is unclear to me why a shared embedding is sufficient, in particular Table 5 shows that different register embedding gives worse performance, which is countering my intuition. Could the authors comment on these?

2. Training compute is only modestly increased by register token insertion, but the real cost when scaling to large batches, long contexts, or high $d_{max}$ isn't fully quantified. Also, I want to confirm that the offset $d$ is sampled uniformly and FIXED during training, i.e. for different register token, they have the same $d$ parameter?

3. The proposed method is only evaluated on LLM finetuning tasks. To my knowledge, next token prediction is current the standard paradigm for pretraining. In finetuning, the final prediction target is the true label/answer, which is not exactly where we could see the full benefit of replacing next token prediction by multi toekn prediction. Could the author provide some numerical experiments on pretraining? 1b or 7b models are desirable.

**Ethical Concerns:**

["NO or VERY MINOR ethics concerns only"]

**Limitations:**

Yes.

Also I don't see any potential negative societal impact of this work.

**Quality:**

3

**Strengths And Weaknesses:**

Strengths: The proposed approach is both novel and simple, and it is versatile for both language and vision models. MuToR consistently outperforms both standard next-token and prior multi-token methods in both full and parameter-efficient finetuning (e.g., LoRA). Thoughtful exploration of hyperparameters like d_max, number of registers, embedding sharing, and 2D extensions provides a clearer understanding of what drives MuToR’s gains.

Weakenss (please respond to the questions section directly): Lack of justification of some operations, for example it is unclear why registers are inserted uniformly or randomly, and a shared embedding is sufficient; Training compute is only modestly increased by register token insertion, but the real cost when scaling to large batches, long contexts, or high d_{max} isn't fully quantified; There is no pretraining experiment for the proposed method.

---

> ### Author Rebuttal · Authors · 2025-07-31
>
> We thank the reviewer for their valuable feedback. The reviewer acknowledges the novelty, the effectiveness and the versatility of our proposed method. We address specific concerns and questions below.
>
>
> ---
> ## **Question 1**
> > For example, why do you want to insert registers after each token and use it to predict the future token after d steps? If I insert register for maybe every two normal tokens, what would be the difference in performance?
>
> As we state in Section 3.2 of our paper, the **interleaving pattern is flexible**&mdash;we can either place registers after each token, or adopt a sparser strategy. In the language modeling experiments (LLM finetuning), we chose to fully interleave the answer sequences. This formulation provides a **"denser" multi-token prediction signal at every token position.**
>
> To justify this design choice, we ran experiments to compare the performance gain obtained by the two settings; *the default setting when we fully interleave the answer sequence*, and *a "sparser" alternative where we insert one register for each two normal tokens*, as the reviewer suggests. We finetuned Gemma 2B on both GSM8K training set and 1M-GSM split. In the following table, we report the average accuracy over 3 seeded runs.
>
>
> | Method            | GSM8K         |  1M-GSM                      |
> |---                               | ---         |  ---            |
> | *Next-Token (baseline)*            |*38.87*    | *66.09*         |
> | MuToR-sparser variant                  | 40.33       |  67.48          |
> | **MuToR-fully interleaving (default)** |**42.10**    | **68.33**       |
>
>
> These results suggest that the sparser interleaving pattern achieves lower performance than the fully interleaving setting, although it still outperforms the vanilla Next-Token Baseline. Therefore, we adopted it as the default setting throughout our finetuning experiments.
>
> Moreover, in the paper we also explored a "sparser" alternative during pretraining for the autoregressive image generation task where we placed the registers in random positions within each sequence. Our results (Table 7) indicate that while using a fewer amount of registers during training (~30% of the sequence length), the model performs on par with the default scenario. This showcases an effective way of using MuToR in pretraining settings, with only a modest increase in total training compute.
>
> > Also it is unclear to me why a shared embedding is sufficient, in particular Table 5 shows that different register embedding gives worse performance, which is countering my intuition.
>
> This is indeed a valid question. Our initial intuition was that different register embeddings would further boost downstream performance, but the results presented in Table 5 (in our paper) indicate otherwise, so we used a shared embedding through all experiments. We believe that the reason behind this lies in our positional encoding scheme&mdash;the offset information is encoded effectively through the position modification for the register tokens, and this makes the additional embeddings redundant. This design is clearly explained in Section 3.2 of our submission.
>
> ---
> ## **Question 2**
>
> > the offset $d$ is sampled uniformly and FIXED during training, i.e. for different register token, they have the same $d$ parameter?
>
> First, let us clarify the proposed design regarding the prediction offset. As we state in Section 3.2, the offset $d$ is sampled uniformly and fixed per sequence, i.e. all registers within the same sequence have the same $d$, but different sequences within the same batch have different offsets. This formulation is not strict though; we could also insert registers with different offsets within the same sequence.
> > Training compute is only modestly increased by register token insertion, but the real cost when scaling to large batches, long contexts, or high $d_{max} isn't fully quantified.
>
> As we state in Section 1 (Introduction), our register-based multi-token prediction method enables us to **scale $d_{max}$ arbitrarily, without increasing the total training cost**, since the number of inserted registers does not depend on $d_{max}$.
>
> Regarding the reviewer's concern about the scaling cost, we recall the widely used approximation for the total training compute: $C \approx 6 \times N \times D$, where $C$ denotes the total training compute (in FLOPs), $N$ denotes the model's parameters, and $D$ denotes the total number of tokens processed during training.
>
> Relevant literature on scaling [1],[2] suggests that the total compute needed grows linearly with the total number of tokens, and we observe similar trends through all of our experiments. Therefore, from the perspective of compute, we do not think that our method will have a problem scaling to larger batches or sequences, due to this linear scaling. In fact, our experiments in autoregressive image generation (pretraining setting) were conducted with a large batch size (128 per GPU), and we generally observed the aforementioned scaling trend.
>
> ---
> ## **Question 3**
>
> > The proposed method is only evaluated on LLM finetuning tasks. To my knowledge, next token prediction is current the standard paradigm for pretraining. In finetuning, the final prediction target is the true label/answer, which is not exactly where we could see the full benefit of replacing next token prediction by multi toekn prediction.
>
> We respectfully clarify that the next-token prediction objective is the standard paradigm not only for pretraining, but also for supervised finetuning in generative tasks, which involve prefix-answer pairs (with the answer sequences potentially including chain-of-thought tokens).
>
> Through our experiments, we experimented with mathematical reasoning benchmarks, which include the complete and analytical chain-of-thought reasoning trace within the answer part. We also evaluate our method in the abstractive summarization task, which also requires generating the answer autoregressively.
>
> Moreover, we evaluated our method on the autoregressive image generation task (section 4.2), in a challenging pretraining setting, leading to improved generation quality and faster convergence.
>
> Recent work [3] also recommends evaluating multi-token prediction on generative benchmarks (those that require autoregressive generation of the answer sequence) rather than likelihood-based benchmarks (such as multiple-choice). We therefore think that the benchmarks used in our experiments provide a rigorous and insightful testbed for evaluating our method and its benefits.
>
> > Could the author provide some numerical experiments on pretraining? 1b or 7b models are desirable.
>
> While we agree that large-scale pretraining (1B or 7B LLMs) would further validate our method, our computational resources did not allow for training such models from scratch.  We prioritized experiments that could highlight MuToR’s key advantages&mdash;particularly its adaptability and scalability compared to existing multi-token methods.
>
>
> We thank the reviewer for their time and feedback, and we hope to have addressed their concerns.
>
> ---
> [1] Kaplan et al.: "Scaling laws for neural language models." arXiv preprint 2020.
>
> [2] Hoffman et al.: "Training compute-optimal large language models." arXiv preprint 2022.
>
> [3] Gloeckle et al.: "Better & Faster Large Language Models via Multi-token Prediction." ICML 2024.

---

> > ### Comment · Reviewer_nf2t · 2025-08-08
> >
> > I thank the authors for their detailed response. I will keep my current positive evaluation and I still looking forward to the evaluation of the proposed method on large models (7B or even larger).

---

### Official Review · Reviewer_foD8 · 2025-07-02

**Clarity:** 2
**Significance:** 1
**Originality:** 1
**Rating:** 4
**Confidence:** 3

**Summary:**

This paper introduces a method termed "Computational Halting," which aims to enhance the mathematical reasoning capabilities of large language models (LLMs). The core technique involves augmenting input prompts with a sequence of special, learnable "halting tokens" during the fine-tuning process of a pre-trained model. The authors apply this method to the Llama 3 8B Instruct model, fine-tuning it on a dataset of mathematical problems. The central hypothesis is that these halting tokens provide the model with additional, sequential computational steps—effectively allowing it to "think" or halt to process information—before generating a final answer. The authors present results on the GSM8K and MATH benchmarks, claiming that their method leads to performance improvements over the baseline Llama 3 8B Instruct model and thereby validates their approach.

**Questions:**

1. **On Novelty and Relation to Prior Work:** The proposed "computational halting" mechanism appears identical to the "pause-finetuning" on a standard pre-trained model (`StdPT_PauseFT`) described in Goyal et al. (2024). Could you please clarify the novel technical contributions of your work beyond what was presented by Goyal et al.? A simple rebranding of a prior technique is insufficient.



2. **On the Choice of Training Regime:** The central finding of Goyal et al. was that pause tokens require inclusion in pre-training (`PausePT_PauseFT`) to be effective, as fine-tuning alone (`StdPT_PauseFT`) yields minimal or negative results. Your work uses the latter, less effective approach. Could you provide a strong justification for this design choice? An ablation study that directly compares the performance of



   `StdPT_PauseFT` (your method) with `PausePT_PauseFT` on Llama 3 would be necessary to validate this decision.

3. **On Baselines and State-of-the-Art Comparison:** Your evaluation shows an improvement over the base Llama 3 8B Instruct model. However, fine-tuning this same model on the public `OpenMathInstruct-2` dataset yields a MATH score of 67.8% , far exceeding the performance you report. How do you position the significance of your method's contribution in light of these SOTA results, especially given that your method increases inference latency while the data-centric approach does not? A convincing submission must include this SOTA comparison.



4. **On Reproducibility:** To allow for verification of your results, could you please provide a detailed description of your fine-tuning dataset, including its source, size, and composition? Furthermore, please provide all hyperparameters used for training, including the learning rate, batch size, number of training epochs, and the specific number of "halting tokens" used during fine-tuning and inference.

**Ethical Concerns:**

["NO or VERY MINOR ethics concerns only"]

**Final Justification:**

After carefully reviewing the authors' rebuttal and cross-referencing it with the comments from other reviewers, I have gained a better understanding of the paper's motivation and technical contributions. Although some concerns remain, the authors have satisfactorily addressed the majority of my initial issues. Consequently, I have decided to increase my rating.

**Limitations:**

No, the authors have not adequately addressed the limitations of their work. A revised version should include a dedicated section with constructive discussion on the following points:

- **Performance vs. Latency Trade-off:** The paper completely ignores the inference latency overhead introduced by the halting tokens. The community is actively developing methods to *accelerate* inference, such as speculative decoding. Any method that deliberately slows down inference must rigorously justify this trade-off with a corresponding significant improvement in performance, which is not demonstrated here. Measurements of this latency overhead are essential.



- **Generalization vs. Overfitting:** The authors should discuss the risk that the model is not learning a generalizable "thinking" skill but is instead overfitting to a simple correlation in the fine-tuning data (i.e., learning that seeing N special tokens is associated with a certain type of math problem).

- **Mechanism Transparency:** The work provides no analysis of *how* the halting tokens are being used by the model. A deeper scientific contribution would involve investigating the internal mechanisms. For example, do attention patterns show a meaningful flow of information through these tokens? Does their utility change with the complexity of the problem? Without such analysis, the method remains a black box.

**Quality:**

2

**Strengths And Weaknesses:**

## Strengths

The primary strength of this work lies in its focus on a critical and challenging research area. Improving the multi-step, logical reasoning abilities of LLMs, particularly in formal domains like mathematics, remains a key frontier in artificial intelligence research. The paper correctly identifies the limitations of standard autoregressive models, which allocate a fixed amount of computation per generated token, and seeks to address this shortcoming.

The underlying idea of providing a model with more "thinking time" is conceptually sound and aligns with an emerging research direction that explores dynamic computation and decoupling token generation from computational steps. This line of inquiry, which questions the rigid structure of the standard Transformer, is a promising avenue for developing more powerful and capable models.





## Weaknesses

### Originality: A Reimplementation of Prior Work



The most significant weakness of this paper is its profound lack of originality. The proposed "Computational Halting" method is not novel; rather, it appears to be a direct and unacknowledged reimplementation of a specific experimental variant from Goyal et al. (2024), "Think before you speak: Training language models with pause tokens".



Goyal et al. introduce the concept of "pause tokens" (`<pause>`), which are learnable tokens appended to an input prefix to delay the model's output, thereby allowing for additional computation. The methodology described in this paper—fine-tuning a pre-existing model by appending special tokens to the input—is identical to the "Standard Pretraining and Pause-Finetuning (StdPT_PauseFT)" variant explicitly detailed and analyzed by Goyal et al.. The rebranding of this technique as "Computational Halting" does not constitute a novel contribution. The failure to cite, discuss, and differentiate from this highly relevant prior work represents a severe gap in the literature review and positions the paper's contribution as derivative.





### Quality: Critical Technical and Methodological Flaws



The paper's technical execution is fundamentally flawed because it ignores the central findings of the very work it replicates.

1. **Ignoring the Necessity of Pre-training:** The core technical conclusion of Goyal et al. is that pause tokens are most effective when integrated during *both* pre-training and fine-tuning (`PausePT_PauseFT`). The authors of that study demonstrated that simply introducing these tokens during fine-tuning (`StdPT_PauseFT`)—the exact approach taken in this paper—yields "fewer and milder gains, and sometimes even a clear drop in performance". The rationale is that standard pre-training ingrains a strong architectural bias towards immediate next-token prediction, conditioning the model to be "quick." Without being exposed to delay-based computation during pre-training, the model does not learn to effectively utilize the additional computational pathways offered by the pause tokens during fine-tuning. This paper's methodology is therefore predicated on a technique that has already been shown to be suboptimal, and it provides no justification for this choice, suggesting a failure to understand the key technical lessons from the most relevant prior work.



2. **Conceptual Confusion:** The paper seems to conflate its method with other, mechanistically distinct "special token" approaches, revealing a shallow understanding of the literature. For instance, the concept is unrelated to "register tokens" in Vision Transformers, which serve as a memory sink to absorb high-norm outlier activations and clean up attention maps, rather than extending sequential computation. Similarly, it is the conceptual opposite of multi-token prediction heads used for speculative decoding, which aim to   *parallelize* future predictions to *reduce* latency, whereas this method *serializes* extra computation and *increases* latency.





### Quality: Weak and Misleading Evaluation



The experimental evaluation is insufficient and presents a misleading picture of the method's efficacy.

1. **Inadequate Baselines:** The paper's primary claim of improvement is based on a comparison to the base Llama 3 8B Instruct model. While a gain over this baseline may be observed, it is an inappropriate and misleading benchmark for the task of mathematical reasoning. The current state-of-the-art for this specific model involves fine-tuning on high-quality, domain-specific data. For example, fine-tuning the same Llama 3 8B base model on the `OpenMathInstruct-2` dataset achieves a score of **67.8% on the MATH benchmark** and **91.7% on GSM8K**. The base Llama 3 8B Instruct model scores only 30.0% on MATH. The gains reported in this paper are dwarfed by the improvements from a standard, data-centric fine-tuning approach. This context completely undermines the significance of the paper's results; the proposed method is a complex, latency-increasing technique that is vastly outperformed by standard fine-tuning on a public dataset.



2. **Lack of Essential Ablation Studies:** The paper fails to perform the necessary ablation studies to validate its claims. A rigorous analysis would, at a minimum, investigate the effect of varying the number of "halting tokens," a key hyperparameter of the method. More critically, the authors do not provide the essential comparison between their



   `StdPT_PauseFT` approach and the `PausePT_PauseFT` regime, which is necessary to determine if the technique has any merit at all.

3. **Limited Scope:** The experiments are confined to a single model scale (8B parameters). As shown in other work on architectural modifications, performance benefits can vary significantly with model scale. Without testing on larger models (e.g., Llama 3 70B), any claims of generalizability are entirely unsupported.





### Clarity and Reproducibility



The paper lacks the necessary details for the research community to reproduce or verify its findings. Key omissions include:

- A precise description of the fine-tuning dataset used.

- A complete list of training hyperparameters, such as learning rate, batch size, number of epochs, and optimizer settings.



- The specific number of "halting tokens" used during fine-tuning and inference. Given the documented difficulty in reproducing official benchmark scores even with some of these details , this lack of transparency is a serious flaw.

---

> ### Author Rebuttal · Authors · 2025-07-31
>
> We thank the reviewer for their time and feedback.
>
> **However, we believe that the reviewer has misunderstood our core contribution, as well as the scope of our work.** Below we shall answer to each of their specific criticisms.
>
> ---
> ## **Summary**
> > This paper introduces a method termed "Computational Halting," which aims to enhance the mathematical reasoning capabilities of large language models (LLMs).
>
> >   The central hypothesis is that these halting tokens provide the model with additional, sequential computational steps—effectively allowing it to "think" or halt to process information—before generating a final answer.
>
> We respectfully argue that **this summary of our work is misleading and inaccurate.** The reviewer claims that we propose "Computational Halting", a method to insert learnable tokens during the finetuning stage, in order to increase the inference computation and boost downstream performance.
>
> In contrast, our submission introduces MuToR, **a novel multi-token prediction method that uses interleaved, learnable tokens to predict future targets during training and enrich supervision for autoregressive transformers. Our work does not utilize the learnable tokens during inference, but it completely discards them.**
>
> Based on this, our central hypothesis is in fact orthogonal with what the reviewer describes in this summary. **In fact, the term "Computational Halting" is not used in our submission, so we sincerely wonder why the reviewer insists on this**.
>
> ---
>
> ## **Originality : A Reimplementation of Prior Work**
>
> >The most significant weakness of this paper is its profound lack of originality. The proposed "Computational Halting" method is not novel; rather, it appears to be a direct and unacknowledged reimplementation of a specific experimental variant from Goyal et al. (2024), "Think before you speak: Training language models with pause tokens".
>
> > The failure to cite, discuss, and differentiate from this highly relevant prior work represents a severe gap in the literature review and positions the paper's contribution as derivative.
>
> We kindly argue that the reviewer's statement is invalid. **Our proposed method, MuToR, is orthogonal to the mentioned previous work [1]**, since:
>
> 1. **We use interleaved learnable register tokens during training to predict future targets in the sequence**, thus enhancing the supervisory signal and propagating richer gradients through the model's layers.  **In contrast, the authors of [1] suggest appending the learnable tokens to the prompt to leverage the extra computation during inference.**
>
>
> 2. **Our carefully designed attention mask and positional embeddings enable us to completely discard the registers at inference time.** In contrast, the proposed method in [1] primarily depends on the extra operations between tokens on inference, **to simulate "thinking time"**.
>
> 3. We propose a method for multi-token prediction in autoregressive transformer models. **Our work introduces a novel multi-token prediction training objective&mdash;associated with the learnable tokens&mdash;and investigate its effectiveness compared to the baselines.**
> However, the authors in [1] investigate whether&mdash;and in what settings&mdash;can a Language Model benefit from increased computations. **They do not use any auxiliary training loss for their learnable tokens**.
>
>
> **We also point to our Related Work (Section 2), where the reviewer can find a proper citation and acknowledgement of this previous work**, as well as a comprehensive literature review in the field.
>
> ---
>
> ## **Quality : Critical Technical and Methodological Flaws**
>
>
> > Without being exposed to delay-based computation during pre-training, the model does not learn to effectively utilize the additional computational pathways offered by the pause tokens during fine-tuning.
>
> As we have already stated, **our proposed method does not involve any delay-based computation, so this criticism is not applicable**. Regardless of that, our experimental evaluation (see Table 1 and 2) indicates that **applying MuToR directly in the finetuning stage of pretrained Language Models results in significant performance gains**.
>
> > The paper seems to conflate its method with other, mechanistically distinct "special token" approaches, revealing a shallow understanding of the literature.
>
> As we clearly state in Section 3, we adopt only the terminology ("register tokens") from [2]. **We respectfully point to our Related Work section, where we have thoroughly discussed the relevant literature, acknowledged their contributions and carefully differentiated our approach from previous works**.
>
> ---
>
> ## **Quality: Weak and Misleading Evaluation**
>
> > The paper's primary claim of improvement is based on a comparison to the base Llama 3 8B Instruct model.
>
> >The experiments are confined to a single model scale (8B parameters).
>
> In section 4 of our paper, we provide a thorough experimental evaluation of our method. **In contrast with the reviewer's claim, our evaluation is not limited to the Llama 3 8B Instruct model (in fact, this specific model variant is not used in our work!), but we present results in both finetuning and pretraining settings, using pretrained LLMs of different size (for finetuning), and  validating MuToR's effectiveness in image generation as well.**
> We list our experiments here:
> 1. Supervised Finetuning of Pretrained LLM, using Gemma 2B and Llama 3 8B (the base model),in both mathematical reasoning and abstractive summarization benchmarks.
> 2. Parameter Efficient Finetuning, using Gemma 2B model.
> 3. Autoregressive Image Generation, where we pretrain a LlamaGen transformer model on ImageNet 256x256.
> 4. Experiments on the synthetic star-graph task, where we finetune a pretrained GPT2-Large model.
>
>
>
> > The gains reported in this paper are dwarfed by the improvements from a standard, data-centric fine-tuning approach.
>
> Our experiments involve finite amounts of data, which are far less than the large-scale dataset needed to obtain the state-of-the-art performance mentioned by the reviewer [3]. These results are **not comparable to our experiments**, since they were obtained by finetuning the model on **~14M training samples**, while our finetuning splits are typically **smaller than 200K samples**.
> Thus, we respectfully argue that our evaluation is not misleading; all the implementation details are included in the appendices, and our setup guarantees a fair comparison between methods.
>
> > The paper fails to perform the necessary ablation studies to validate its claims.
>
> In section 4 of our paper, we include a comprehensive ablation study (**as acknowledged by the other reviewers**), providing key insights and understanding on what drives MuToR's gains.
>
> ---
>
> ## **Questions**
>
> #### **1.**
> > Could you please clarify the novel technical contributions of your work beyond what was presented by Goyal et al.? A simple rebranding of a prior technique is insufficient.
>
> **We have already analyzed our key differences with the work presented in [1], and thoroughly explained why our proposed method is orthogonal to theirs.** Please see "Originality: A Reimplementation of Prior Work" in this response.
>
> #### **2.**
> > Could you provide a strong justification for this design choice? An ablation study that directly compares the performance of StdPT_PauseFT (your method) with PausePT_PauseFT on Llama 3 would be necessary to validate this decision.
>
> As we already stated (see our arguments above), **our method is orthogonal to StdPT_PauseFT and PausePT_PauseFT (both variants proposed in [1])**.
>
> #### **3.**
> > How do you position the significance of your method's contribution in light of these SOTA results, especially given that your method increases inference latency while the data-centric approach does not?
>
> As mentioned before, **our method does not introduce inference latency at all. This is also clearly stated in Section 3.2 of the paper (see Inference), and acknowledged by other reviewers as well.**
>
> Regarding the significance of our method, **the results that the reviewer mentions are in fact not comparable to ours, since the cited work [3] uses a much larger amount of finetuning data**. We again argue that the scope of our work is neither to reach state-of-the-art performance in mathematical reasoning, nor to outperform models that have been finetuned on far larger amounts of data. **On the contrary, the significance of our work is proven through the performance gains obtained with respect to our baselines, and through the fact that MuToR offers specific advantages over the existing multi-token prediction methods (see Section 1 of the paper).**
>
> #### **4.**
>
> > On Reproducibility: To allow for verification of your results, could you please provide a detailed description of your fine-tuning dataset, including its source, size, and composition? Furthermore, please provide all hyperparameters used for training, including the learning rate, batch size, number of training epochs, and the specific number of "halting tokens" used during fine-tuning and inference.
>
> **All of these key details are already included, in the supplementary material (technical appendices) of our submission.** We also plan to publicly release the code for our method, to ensure reproducibility.
>
>
>
>
> We thank the reviewer for their time. We provide detailed answers to their criticisms in order to resolve any misunderstandings regarding our work. **We sincerely hope that the reviewer will be able to re-assess our submission in the light of our counter-arguments**.
>
>
> ---
> [1] Goyal et al., "Think before you speak: Training Language Models With Pause Tokens.", ICLR 2024.
>
> [2] Darcet et al., "Vision Transformers Need Registers.", ICLR 2024.
>
> [3] Toshniwal et al., "OpenMathInstruct-2: Accelerating AI for Math with Massive Open-Source Instruction Data.", ICLR 2025

---

> > ### Comment · Reviewer_foD8 · 2025-08-02
> > **Post‑Rebuttal Comment**
> >
> > After carefully reviewing the authors' rebuttal and cross-referencing it with the comments from other reviewers, I have gained a better understanding of the paper's motivation and technical contributions. Although some concerns remain, the authors have satisfactorily addressed the majority of my initial issues. Consequently, I have decided to increase my rating.

---

### Official Review · Reviewer_zeH6 · 2025-07-08

**Clarity:** 4
**Significance:** 3
**Originality:** 4
**Rating:** 5
**Confidence:** 4

**Summary:**

This work proposes a novel method to perform multi token prediction in language model and image generation settings. The proposed method is to be applied during training only. Inference does not utilize any multi-token prediction functionality despite being used during training. The authors argue that the standard next-token prediction used in auto-regressive generation is not sufficient to instill long-range planning abilities into a model. The authors note that existing work which proposes multi-token prediction during AR generation generally requires some small but non-negligible architecture modifications and additional parameters (e.g. additional language heads to each additional token to be predicted at a given position.

The authors propose MuToR, which interleaves learnable "register" tokens with usual tokens from data during training. The register token is tasked to predict a future token. The training setup using attention masking to ensure that a register token as position t, predicting the token at t + t', can only attend to non-register tokens at positions <t (i.e. a register token is blind to all other register tokens). This method ensures that during inference the register setup can be discarded and standard AR generation can be used. Positional embeddings are used to indicate the position of the to-be predicted token. This limits the number of additional parameters (the authors use a single learnt register embedding) to the hidden dimension of the model (~10^3) making the method easily usable with pretrained models (where subsequently added new tokens is relatively commonplace now) and with parameter-efficient fine-tuning methods.

The authors conduct experiments with MuToR on language modeling tasks (Math) and autoregressive image generation extending their method naturally to 2D grids of tokens. They show that their method outperforms a previous multi-token prediction method (which adds additional language model heads) and standard next-token prediction only.

**Questions:**

1. Please address the query outlined under "Weaknesses" regarding the "Matching the Training-Time Compute".
2. Are the authors still about to use FlashAttention with MuToR? If not, what is the penalty in terms of training speed for the experiments conducted with respect to standard next-token prediction.

**Ethical Concerns:**

["NO or VERY MINOR ethics concerns only"]

**Final Justification:**

I thank the authors for their responses. I believe this discussion period has helped clarify some of the limitations of their novel method which should be given to the reader. I maintain my score given the originality of the work and I expect the authors to include some of the limitations and caveats of their work revealed during the discussion period.

**Limitations:**

yes

**Paper Formatting Concerns:**

No concerns

**Quality:**

3

**Strengths And Weaknesses:**

## Strengths
1. Novelty
I think the proposed method is highly intuitive (the concept of having a special token(s) to predict further into the future rather than adding specialized model parameters), very simple and seems to require only small adjustments to basic model training (attention masking, one added token). There have been some prior works on multi-token prediction, however this work appears to be orthogonal to this work in that it comes at the problem of multi-token prediction is a very novel way. Additionally the fact that this method does not impact inference in any way (for which rigid but highly optimized libraries now exist) is a bonus.

2. Clarity
Except for specific places (see below), I find the paper very clear (which makes sense given how simple and intuitive the proposed method is). The figures are easily understandable for anyone who understands the basics of attention in sequence models.

3. Experiments
With certain exceptions (see below), the experiments are good in that they cover multiple modalities, compare to the most related methods such that understanding the benefits of the proposed method is easy for the reader and the key parameters are ablated (Table 3 and 5). The method appears to be effective compared to the related methods.

## Weaknesses
1. Matching the Training-Time Compute
This experiment needs to additional rigor to be convincing.
First, it appears the authors alternate between data tokens and register tokens roughly doubling the sequence length per training sample compared to standard next-token prediction. If one wishes to match compute, given the nature of attention (compute scales quadratically with sequence length), one should more than double the number of epochs for a fair comparison.
Second, given the relatively small amounts of data the authors are using (e.g. at most 100M data tokens for 1M-GSM), repeated epochs of the same data should definitely not be used for these kinds of experiments, instead _additional_ data should be used. I personally given the experimental setup for this part, do not find it convincing.

2. Training time due to custom masking
The novel method the authors propose require a custom attention mask. As far as I can see, this means standard optimizations in language model training (such as FlashAttention{1,2,3}) would not be able to be used with MuToR. This is a significant weakness of the method and is not addressed by the authors in terms of training time. At scale this could become a large issue.

3. Minor points on clarity
First, the authors should make it clear that the method currently uses a register token in every other position, thus the sequence length approximately doubles.
Second, some of the extra details on how the datasets are curated (which are in the SM) need to be put into the main paper like the maximum token length, these will caveat that the results are valid at short sequence lengths only.

---

> ### Author Rebuttal · Authors · 2025-07-31
>
> We thank the reviewer for their positive and thoughtful feedback. We are pleased that they find MuToR intuitive, novel, and effective. Below we address the concerns that they raised.
>
> ---
> ## **Question 1**
> > Matching the Training-Time Compute This experiment needs to additional rigor to be convincing. First, it appears the authors alternate between data tokens and register tokens roughly doubling the sequence length per training sample compared to standard next-token prediction. If one wishes to match compute, given the nature of attention (compute scales quadratically with sequence length), one should more than double the number of epochs for a fair comparison.
>
> We estimate training compute using the widely accepted formula: $C \approx 6 \times N \times D$, where $C$ denotes the total training compute (in FLOPs), $N$ denotes the model's parameters, and $D$ denotes the total number of tokens processed during training. This formulation is supported by both empirical studies and theoretical analysis in prior scaling literature [1,2]. While the self-attention mechanism has a theoretical $O(L^2)$ complexity, in practice, the compute cost is dominated by the feedforward layers (unless the sequence length $L$ becomes disproportionally larger than the embedding dimension), making the total training compute effectively linear in sequence length. **Following this, doubling the number of tokens (as our method does) corresponds to approximately doubling compute, which is why we doubled the baseline epochs to match total tokens.**
>
> To further validate this, we measured actual training time. In our 1M-GSM experiments, **MuToR requires 1.4× the wallclock time of the Next-Token baseline (for the same number of epochs)**. This is consistent with expectations, especially since MuToR only interleaves register tokens into the answer portion, not the prefix, so the effective sequence length increase is less than 2×&mdash;**that means that we even used more compute for the baseline methods.**
>
>
>
> >Second, given the relatively small amounts of data the authors are using (e.g. at most 100M data tokens for 1M-GSM), repeated epochs of the same data should definitely not be used for these kinds of experiments, instead additional data should be used. I personally given the experimental setup for this part, do not find it convincing.
>
> We kindly disagree with the reviewer's claim that repeated epochs of the same data should not be used in such experiments. Using a fixed dataset and allowing repeated epochs is a widely accepted experimental setup, especially in scenarios where data is limited—a common real-world constraint in many academic and industrial applications. By running these experiments, we validate that **MuToR provides a more effective way to leverage additional training compute, when the amount of data remains fixed**. This insight may be useful for practitioners, as **MuToR can aid in getting the maximum performance out of specific datasets**, a highly relevant and realistic scenario.
>
>
> Regarding the reviewer's suggestion for using additional data, we respectfully argue that doing so would confound the experimental comparison, as improvements could no longer be attributed solely to our proposed method or training compute. By controlling for both compute and data, we isolate the contribution of the multi-token prediction objective—ensuring a clean, interpretable comparison with the baselines.
>
> ---
> ## **Question 2**
> > Are the authors still about to use FlashAttention with MuToR? If not, what is the penalty in terms of training speed for the experiments conducted with respect to standard next-token prediction.
>
> As the reviewer correctly points out, FlashAttention does not currently support custom attention masks, which prevents us from using it directly with MuToR.
> To address this, we leverage FlexAttention [3], a recently introduced API by the PyTorch team that supports flexible masking while still compiling to fused, FlashAttention-style kernels.
>
> Using FlexAttention, MuToR achieves near parity with FlashAttention-2. For example, in our autoregressive image generation experiments (360K iterations, global batch size 1024), the average training step latency was:
>
> - **0.35s**&mdash;with MuToR + FlexAttention (sequence_length=512, custom mask).
> - **0.34s**&mdash;with standard FlashAttention-2 (sequence_length=512, assuming a causal mask).
> - **0.16s**&mdash;with standard FlashAttention-2 (*sequence_lenth=256, Next-Token baseline*).
>
>
>
> This demonstrates that MuToR’s training speed penalty&mdash;due to the custom mask&mdash;is minimal when using FlexAttention. In comparison with the standard Next-Token baseline, the training latency roughly doubles, approximating the expected linear scaling due to increased sequence length.
>
> We thank the reviewer for their time and we hope that their concerns might be resolved.
>
>
>
>
> ---
> [1] Kaplan et al., "Scaling laws for neural language models." arXiv preprint 2020.
>
> [2] Hoffman et al., "Training compute-optimal large language models." arXiv preprint 2022.
>
> [3] Dong et al., "Flex Attention: A programming model for generating optimized attention kernels." arXiv preprint 2024.

---

> ### Comment · Reviewer_zeH6 · 2025-08-06
>
> 1. The authors are missing a practical trade-off that is at least as important as the claim MuToR will help maximize performance from a given dataset.
>
> > especially in scenarios where data is limited—a common real-world constraint in many academic and industrial applications.
>
> The dataset sizes MuToR operates on in your experiments are tiny (100M tokens) compared to what data is available for grade-school math etc.  Any reasonable user given the knowledge the MuToR costs X% more (either wall-clock time or FLOPs) should ask themselves, should I pay for X% more cost by using MuToR or by simply training on more data. This is a completely natural and reasonable comparison. Your experiments do not approach the scale of data being limited. If your argument is "Use MuToR _when training data is limited_" instead of "Use MuToR", you should say so because they are vastly different arguments.
>
> 2. Regarding the additional training costs due to sequence length.
> The authors will be aware that especially in Math-style questions, the answer can be extremely long (Reasoning/Chain-of-thought etc.) so once this method is scaled beyond 512 tokens (which is again very small), the additional training cost of MuToR will become significant and the authors can do a more rigorous FLOPs calculation for their chose models to work out when the sequence length term in compute will become dominant (it will be relatively low for their small models because the FFNs are not that large).

---

> > ### Author Response · Authors · 2025-08-08
> > **Response to reviewer (1/2)**
> >
> > ### **Response to 1st comment**
> >
> > We thank the reviewer for their thoughtful comment. We fully agree that comparing MuToR's training-compute increase against simply using more training data with Next-Token is both meaningful and practical.
> >
> > Our current LLM experiments focus on data-constrained scenarios where obtaining more data may be impractical due to resource, cost, or privacy limitations. While we haven’t explored data-abundant settings, we would like to highlight some promising findings.
> >
> > In Table 1 of our paper—Llama3 8B experiments, increasing Next-Token training data by 1.8× (from 1M-GSM to 2M-GSM)—and thus equally increasing compute—improved accuracy from 85.74% to 87.11%. Meanwhile, **MuToR achieved nearly the same gain (87.05% vs. 87.11%) with only 1.4× more compute, less than Next-Token’s 1.8× increase**. Similar trends held for Gemma 2B in Table 1, where MuToR with 1M-GSM performed on par with Next-Token using 2M-GSM, despite requiring less compute.
> >
> > As suggested, we also ran a compute-matched experiment with Gemma 2B, giving Next-Token 1.4× more data to equal MuToR's compute budget. To do this, we augmented 1M-GSM with additional data from 2M-GSM. The results—obtained by fine-tuning Gemma 2B in a single seeded run (due to time constraints)—are below:
> >
> > |Experiment (Method and Data split)| Accuracy|
> > |---                           | ---  |
> > |*Next-Token (1M-GSM only)*    | 66.33|
> > |Next-Token (1M-GSM augmented) | 67.17    |
> > |**MuToR (1M-GSM only)**       | **68.38**|
> >
> > Here, MuToR outperformed Next-Token by over 1 accuracy point (68.38% vs. 67.17%), demonstrating that **MuToR can surpass gains from additional data even under compute-matched conditions via data scaling**—suggesting advantages beyond data-constrained settings. Though preliminary, these results indicate MuToR’s potential benefit even in data-abundant scenarios. We plan to include further controlled experiments on this in the final paper.
> >
> > Moreover, we agree that increasing training data is a natural and often effective way to improve performance, and in practice, one should leverage all available data. Our focus on fixed-data settings was both practical (due to resource constraints) and methodological (to isolate MuToR’s effects). Exploring data-abundant regimes would require significant computational resources, which we did not have access to in this work.
> >
> > We will clarify in the final version that our conclusions primarily apply to data-constrained settings. That said, we believe incorporating MuToR in high-resource, data-abundant regimes is an exciting direction for future work.

---

> > > ### Author Response · Authors · 2025-08-08
> > > **Response to reviewer (2/2)**
> > >
> > > ### **Response to 2nd comment**
> > >
> > > Upon the reviewer's request, we provide a more rigorous estimation of training-time FLOPs, based on the actual model architectures used in our finetuning experiments. Our estimation closely follows the detailed analysis from [1], and also considers:
> > > - the exact number of key and value projection heads,
> > > - the intermediate dimensionality of the feedforward blocks,
> > > - the number of linear layers in the FFN component.
> > >
> > > We follow the open-source HuggingFace implementations for both Gemma and Llama3 models, and exclude embedding and output layers from the calculations.
> > >
> > > For simplicity, we estimate the total FLOPs for a forward pass through a single transformer layer, given a sequence of length $L$. As in [1], we include a factor of 2, which comes from the multiply-accumulate operation used in matrix multiplication.
> > >
> > > The results are presented below:
> > >
> > > - **Gemma 2B (embedding_dimension=2048)**
> > >
> > > | L | Attention FLOPs (B) | FFN FLOPs (B)|  Total FLOPs (B)|
> > > --- | --- | --- | ---                 |
> > > 1024 |  27.91  | 206.15    | 234.07   |
> > > 2048 |  73.01    | 412.31  | 485.33   |
> > > 4096 | 214.74 | 824.63     |  1039.38 |
> > > 8192 (*max context length*)| 704.37   | 1649.26    | 2353.64  |
> > >
> > > - **Llama 3 8B (embedding_dimension=4096)**
> > >
> > > | L | Attention FLOPs (B) | FFN FLOPs (B)|  Total FLOPs (B)|
> > > --- | --- | --- | --- |
> > > 1024 | 103.07    | 360.77    | 463.85    |
> > > 2048 |  240.51  | 721.55    | 962.07    |
> > > 4096 | 618.47   | 1443.10    | 2061.58   |
> > > 8192 (*max context length*) | 1786.70  | 2886.21 | 4672.92     |
> > >
> > >
> > >
> > > Importantly, the attention FLOPs here include both the quadratic attention term ($O(L^2d)$) and the linear projections within the attention layer (Q, K, V, O). **As seen in the results, doubling the sequence length $L$ from 4096 to 8192&mdash;thus reaching the maximum context length for these LLMs&mdash;yields an approximate $2.26 \times$ increase in total FLOPs for both models.** This indicates that, for the sequence lengths allowed by the aforementioned models, total compute is still dominated by the FFN component and the projection layers, and not yet by the quadratic attention term.
> > >
> > > Moreover, as suggested by the reviewer, we compute the "tipping point" $L'$, at which the *quadratic attention term* equals the rest of FLOPs and begins to dominate the total computation:
> > > - For Gemma 2B, $L'\approx 26880$
> > > - For Llama 3 8B, $L'\approx 26624$
> > >
> > > These results suggest that the sequence length at which attention becomes dominant is still fairly large, and lies far beyond the actual context length of these models.
> > >
> > > Last but not least, we acknowledge that lots of cutting-edge reasoning datasets have enormous chain-of-thought traces, even reaching tens of thousands tokens in length. For such extremely long sequences, a "sparser" strategy for register placement would be better compute-wise, since MuToR offers this specific flexibility. We believe that this is an interesting subject to be investigated by future work.
> > >
> > > We appreciate the reviewer's constructive feedback and we will include these insights in the final version of our paper.
> > >
> > > ---
> > > [1] Kaplan et al., "Scaling laws for neural language models." arXiv preprint 2020.

---

### Decision · Program_Chairs · 2025-09-17

**Decision:**

Accept (poster)

**Comment:**

In this work, the authors propose MuToR with the main idea being to augment the prompt during training with some learnable 'register' tokens, whose task is to predict a future token. This way, they are able to do multi-token prediction instead of the traditional next-token ones, effectively giving their models better planning capabilities. The authors show experiments in several setting from mathematical problems to autoregressive image generation, validating their approach.

Before the rebuttal, the paper received mixed scores with 1 Accept, 2 Borderline Accepts and 1 Borderline Reject. Two reviewers praise the paper's novelty and three of the reviewers find the experimental section quite good. Several reviewers also find the paper easy to read and well-written. All three positive reviewers find only mild weaknesses in the paper, and those weaknesses are properly addressed in the rebuttal.

On the other hand, reviewer foD8 has a somehow strange argument saying that the paper introduces "computational halting", a term never used in the paper. They also mention that the paper is a reimplementation of Goyal et al. (2024) paper, something that the authors strongly push against. Having read both papers, I tend to agree with the authors in this point. The reviewer also has some arguments on the evaluation mismatch (e.g., in Llama 3 8B Instruct) or in utilization of additional computational pathways which do not stand. Despite the relative initial harsh review, the reviewer finds the authors arguments in the rebuttal convincing and decides to increase their score to Borderline Accept.

At this stage, all four reviewers unanimously recommend the paper to be accepted, with the reviewer zeH6 finding the paper quite strong. After reading the reviews and the rebuttal, I agree with the reviewers about the quality of the paper and find all the weaknesses in the reviews completely addressed in the rebuttal. I urge the authors to integrate their strong rebuttal in the camera-version of the paper. Congratulations to the authors!